# Emergent Symbols through Binding in External Memory

**Taylor W. Webb**
University of California Los Angeles
Los Angeles, CA
`taylor.w.webb@gmail.com`

**Ishan Sinha, Jonathan D. Cohen**
Princeton University
Princeton, NJ

## Abstract

A key aspect of human intelligence is the ability to infer abstract rules directly from high-dimensional sensory data, and to do so given only a limited amount of training experience. Deep neural network algorithms have proven to be a powerful tool for learning directly from high-dimensional data, but currently lack this capacity for data-efficient induction of abstract rules, leading some to argue that symbol-processing mechanisms will be necessary to account for this capacity. In this work, we take a step toward bridging this gap by introducing the Emergent Symbol Binding Network (ESBN), a recurrent network augmented with an external memory that enables a form of variable-binding and indirection. This binding mechanism allows symbol-like representations to emerge through the learning process without the need to explicitly incorporate symbol-processing machinery, enabling the ESBN to learn rules in a manner that is abstracted away from the particular entities to which those rules apply. Across a series of tasks, we show that this architecture displays nearly perfect generalization of learned rules to novel entities given only a limited number of training examples, and outperforms a number of other competitive neural network architectures.

## 1 Introduction

Human intelligence is characterized by a remarkable capacity to detect the presence of simple, abstract rules that govern high-dimensional sensory data, such as images or sounds, and then apply these to novel data. This capacity has been extensively studied by psychologists in both the visual domain, in tasks such as Raven's Progressive Matrices (Raven & Court, 1938), and the auditory domain, in tasks that employ novel, artificial languages (Marcus et al., 1999).

In recent years, deep neural network algorithms have reemerged as a powerful tool for learning directly from high-dimensional data, though many studies have now demonstrated that these models suffer from similar limitations as those faced by the earlier generation of neural networks: requiring enormous amounts of training data and tending to generalize poorly outside the distribution of those training data (Lake & Baroni, 2018; Barrett et al., 2018). This stands in sharp contrast to the ability of human learners to infer abstract structure from a limited number of training examples and then systematically generalize that structure to problems involving novel entities.

It has long been argued that the human ability to generalize in this manner depends crucially on a capacity for variable-binding, that is, the ability to represent a problem in terms of abstract symbol-like variables that are bound to concrete entities (Holyoak & Hummel, 2000; Marcus, 2001). This in turn can be broken down into two components: 1) a mechanism for *indirection*, the ability to bind two representations together and then use one representation to refer to and retrieve the other (Kriete et al., 2013), and 2) a representational scheme whereby one of the bound representations codes for abstract variables, and the other codes for the values of those variables.

In this work, we present a novel architecture designed around the goal of having a capacity for abstract variable-binding. This is accomplished through two important design considerations. First, the architecture possesses an explicit mechanism for indirection, in the form of a two-column external memory. Second, the architecture is separated into two information-processing streams, one that maintains learned embeddings of concrete entities (in our case, images), and one in which a recurrent controller learns to represent and operate over task-relevant variables. These two streams only interact in the form of bindings in the external memory, allowing the controller to learn to perform tasks in a manner that is abstracted away from the particular entities involved. We refer to this architecture as the Emergent Symbol Binding Network (ESBN), due to the fact that this arrangement allows abstract, symbol-like representations to emerge during the learning process, without the need to incorporate symbolic machinery.

We evaluate this architecture on a suite of tasks involving relationships among images that are governed by abstract rules. Across these tasks, we show that the ESBN is capable of learning abstract rules from a limited number of training examples and systematically generalizing these rules to novel entities. By contrast, the other architectures that we evaluate are capable of learning these rules in some cases, but fail to generalize them successfully when trained on a limited number of problems involving a limited number of entities. We conclude from these results that a capacity for variable-binding is a necessary component for human-like abstraction and generalization, and that the ESBN is a promising candidate for how to incorporate such a capacity into neural network algorithms.

## 2 TASKS

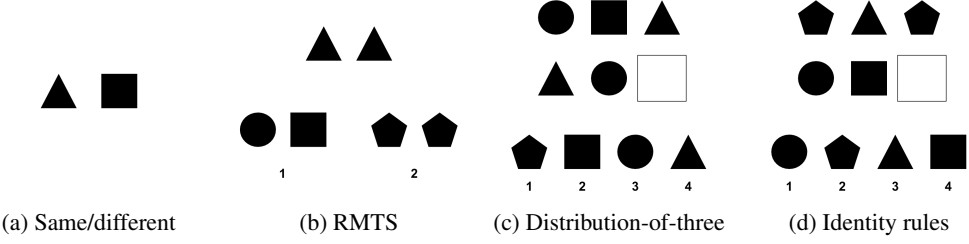

|  (a) Same/different | (b) RMTS | (c) Distribution-of-three | (d) Identity rules |

Figure 1: Abstract rule learning tasks. Each task involves generalizing rules to objects not seen during training. (a) Same/different discrimination task. (b) Relational match-to-sample task (answer is 2). (c) Distribution-of-three task (answer is 2). (d) Identity rules task (ABA pattern, answer is 1).

We consider a series of tasks, each involving the application of an abstract rule to a set of images. For all tasks, we employ the same set of $n = 100$ images, in which each image is a distinct Unicode character (the specific characters used are shown in A.7). We construct training sets in which $m$ images are withheld (where $0 \leq m \leq n - o$, and $o$ is the minimum number of images necessary to create a problem in a given task) consisting of problems that employ only the remaining $(n - m)$ images, and then test on problems that employ only the $m$ withheld images, thus requiring generalization to novel entities. In the easiest generalization regime ($m = 0$) the test set contains problems composed of the same entities as observed during training (though the exact order of these entities differs). In the most extreme generalization regime, we evaluate models that have only been trained on the minimum number of entities for a given task, and then must generalize what they learn to the majority of the $n$ images in the complete set. This regime poses an extremely challenging test of the ability to learn to perform these tasks from limited training experience, in a manner that is abstracted away from the specific entities observed during training.

The first task that we study is a *same/different discrimination* task (Figure 1a). In this task, two images are presented, and the task is to determine whether they are the same or different. Though this task may appear quite simple, it has been shown that the ability to generalize this simple rule to novel entities is actually a significant challenge for deep neural networks (Kim et al., 2018), a pattern that we also observe in our results.

The second task that we consider is a *relational match-to-sample* (RMTS) task (Figure 1b), essentially a higher-order version of a same/different task. In this task, a *source* pair of objects is

compared to two *target* pairs. The task is to identify the target pair with the same relation as the source pair; e.g., if the source pair contains two of the same object, to identify the target pair that contains two of the same object. It was initially believed that the ability to perform this task is not unique to humans (Premack, 1983), but it has now been shown that this ability depends on a visual entropy confound that arises from using large arrays of objects rather than simple pairs (Fagot et al., 2001). When the task is presented in a manner that does not allow this confound to be exploited (as is the case in our experiments), the ability to perform the task with novel entities appears to be unique to humans, and therefore is a good test of the human ability for abstract rule learning.

Next we consider a task based on Raven's Progressive Matrices (RPM; Raven & Court (1938)). RPM is a commonly used visual problem-solving task, and is one of the most widely used tests of *fluid intelligence* (Snow et al., 1984), the ability to reason and make inferences in a novel domain (as opposed to *crystallized intelligence*, the ability to solve familiar tasks). In this task, a $3 \times 3$ array of figural elements is presented, in which the elements are governed by a simple rule, or set of rules, with the lower right element of the array left blank. The task is to infer the rule that governs the elements in the array, and then use that rule to select from among $8$ candidate completions. Many of the rules that govern RPM problems are relations involving sets. One such rule is sometimes referred to as *distribution-of-three* (Carpenter et al., 1990), according to which the same set of three elements (e.g. a triangle, square, and circle) will appear in each row, though the order doesn't matter. The task in this case is simply to identify the set, determining which element is missing from the final row, and locating this element among the choices.

Though multiple RPM-inspired datasets have recently been proposed (Barrett et al., 2018; Zhang et al., 2019), in this work we choose to strip away unnecessary complexity, focusing on $2 \times 3$ arrays governed by a single rule (Figure 1c), in order to focus specifically on the capacity for generalization of an abstract rule to novel entities. We find that, even in this simplified setting, this form of generalization is extremely challenging.

The final task that we consider is a visual version of the *identity rules* task studied by Marcus et al. (1999). In this task, an abstract pattern (e.g. ABA or ABB) must be inferred from a sequence of elements. For instance, in the original study, the following sequence 'ga ni ga, li na li, wo fe wo' is governed by an ABA rule, whereas the sequence 'ga ni ni, li na na, wo fe fe' is governed by an ABB rule. This study played an important role in debates concerning the presence of algebraic rule-like processes in human cognition, because it demonstrated that even 7-month-old human infants are capable of detecting this abstract regularity and generalizing it to novel entities, whereas neural networks tend to overfit to the specific entities involved and fail to generalize the rule.

In our implementation, we use visual images rather than sounds, and present the task as a $2 \times 3$ array (Figure 1d). In this task, each problem is governed by either an ABA, ABB, or AAA rule. The task is to determine which of these patterns is present in the first row, and then to apply that pattern by selecting an element from a set of 4 choices to complete the second row.

For all four tasks, we consider generalization regimes in which some number of images ($m \in \{0, 50, 85, 95\}$ out of $n = 100$) are withheld from training. For the same/different discrimination task, on which only two images are necessary to construct a problem, we also consider the case in which $m = 98$ (such that the training set consists of problems involving only $n - m = 2$ images, the minimum number necessary to construct the task).

In most settings, we construct training sets consisting of $10^4$ problems. This is a tiny fraction of all possible problems (on the order of $10^9$ when the multiple choice options are considered)[1] Thus, even in the easiest generalization regime ($m = 0$) this is an extremely small amount of training data relative to the size of the task space. In the most extreme regimes, in which $m \geq 95$, it is only possible to construct a few hundred problems, resulting in even more limited training experience.

## 3 APPROACH

For each task, we treat the problem as a sequence of images $\boldsymbol{x}_{t=1} \ldots \boldsymbol{x}_{t=T}$, with an associated target $\boldsymbol{y}$. In the same/different discrimination task, there are $T = 2$ images, and $\boldsymbol{y}$ is a binary target indicating whether the images are the same or different. In the RMTS task, there are $T = 6$ images,

---

[1]Except for the same/different task, as detailed in A.2.

consisting of the source pair followed by two target pairs, and $y$ is a binary target indicating which target pair matches the source pair. In both the distribution-of-three task and the identity rules task, there are $T = 9$ images, consisting of the three entries in the first row, the two non-empty entries in the second row, and the four multiple-choice options, and $y$ is a four-way classification target, indicating which of the multiple-choice options is correct.

All images are $32 \times 32$ grayscale images containing a single Unicode character. For each problem, we first process each image independently by a shared encoder $f_e$, generating image embeddings $z_{t=1} \ldots z_{t=T}$, and then pass these embeddings to a sequential model component $f_s$ that generates a response (either through a sigmoid output layer for tasks with a binary target, or a softmax layer for tasks with a four-way classification target). The sequential component is either the ESBN or one of a number of alternative architectures described below. We use the same encoder architecture $f_e$ (detailed in A.3) for all models. All components are trained end-to-end, including the encoder.

## 3.1 TEMPORAL CONTEXT NORMALIZATION

We use temporal context normalization (TCN), recently shown to improve out-of-distribution generalization in relational reasoning tasks (Webb et al., 2020). TCN is similar to batch normalization, but, instead of normalizing over the batch dimension, normalizes over a task-relevant temporal window. This has the effect of preserving information about the relations between the entities present within this window (e.g. the size of those entities relative to one another), resulting in better generalization of learned relations to novel contexts (i.e. out-of-distribution).

We found that TCN significantly improved generalization for all of the models on all of the tasks studied in the present work[2]. Therefore, the primary results we report all incorporate this technique ( A.5.1 includes a comparison of the performance of all models on all tasks with and without TCN). Specifically, we applied TCN to the embeddings $z_{t=1} \ldots z_{t=T}$ extracted by the encoder. Webb et al. (2020) also reported that it is sometimes useful to apply TCN separately to different components of a sequence. We found that this was the case for the RMTS task that we studied, in which we found it useful to apply TCN separately to the embeddings for the source pair and each target pair. For all of the other tasks that we studied, TCN was applied over the entire sequence for each problem.

## 3.2 EMERGENT SYMBOL BINDING NETWORK

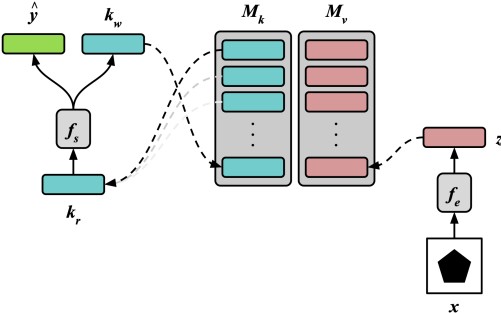

Figure 2: Emergent Symbol Binding Network. $f_s$ consists of an LSTM controller plus output layers for $\hat{y}$, $k_w$, and $g$ (not shown). $f_e$ is a multilayer feedforward encoder that translates an image $x$ into a low-dimensional embedding $z$. These two pathways only interact indirectly via a key/value memory.

The ESBN (Figure 2; Algorithm 1) uses an LSTM controller ($f_s$) with a differentiable external memory that is explicitly separated into keys ($M_k$) and values ($M_v$). At each time step $t$, a key/value pair is written to memory. The keys written to memory, $k_{w_t}$, are generated by an output layer from the LSTM controller, and the values are the individual input embeddings, $z_t$, of the input sequence, unmodified by the LSTM. Our hypothesis was that factoring the model into two separate information processing streams would allow the LSTM to learn how to represent abstract variables in the keys

---

[2]Except for the PrediNet on the same/different task, as detailed in A.5.1.

it generates, which could then be explicitly bound to associated values (image embeddings) learned by the separate encoder network ($f_e$), allowing the ESBN to employ a form of indirection.

To retrieve keys from memory, similarity scores are computed by comparing (via a dot product) the image embedding $z_t$ to all of the values in memory $M_{v_{t-1}}$. These similarity scores are passed through a softmax nonlinearity to generate weights $w_{k_t}$, and passed through a sigmoid nonlinearity (with learned gain and bias parameters, $\gamma$ and $\beta$) to generate confidence values $c_{k_t}$ (one weight and confidence value per entry in memory). The weights are used to compute 1) a weighted sum of all keys in memory $M_{k_{t-1}}$, and 2) a weighted sum of all associated confidence values $c_{k_t}$. Finally, the retrieved key and associated confidence value are concatenated and multiplied by a learned sigmoidal gate $g_t$ to form $k_{r_t}$, the input to the LSTM controller at the next time step.

---

**Algorithm 1:** Emergent Symbol Binding Network. ($\|$) indicates the concatenation of a vector and a scalar, forming a vector with one additional dimension. $\{,\}$ indicates the concatenation of a matrix and a vector, forming a matrix with one additional row. $\sigma()$ is the logistic sigmoid function.

$k_{r_{t=0}} \leftarrow \mathbf{0}$;
$h_{t=0} \leftarrow \mathbf{0}$;
$M_{k_{t=0}} \leftarrow \{\}$;
$M_{v_{t=0}} \leftarrow \{\}$;
**for** $t$ in $1 \ldots T$ **do**

    $z_t \leftarrow f_e(x_t)$;
    $\hat{y}_t, g_t, k_{w_t}, h_t \leftarrow f_s(h_{t-1}, k_{r_{t-1}})$;
    **if** $t$ *is* 1 **then**
        $k_{r_t} \leftarrow \mathbf{0}$;
    **else**
        $w_{k_t} \leftarrow \text{softmax}(M_{v_{t-1}} \cdot z_t)$;
        $c_{k_t} \leftarrow \sigma(\gamma(M_{v_{t-1}} \cdot z_t) + \beta)$;
        $k_{r_t} \leftarrow g_t \sum_{i=1}^{t-1} w_{k_t}(i)(M_{k_{t-1}}(i)\|c_{k_t}(i))$ ;
    **end**
    $M_{k_t} \leftarrow \{M_{k_{t-1}}, k_{w_t}\}$;
    $M_{v_t} \leftarrow \{M_{v_{t-1}}, z_t\}$;
**end**
**return** $\hat{y}_{t=T}$

---

### 3.3 ALTERNATIVE ARCHITECTURES

The simplest alternative architecture that we consider is an *LSTM* (Hochreiter & Schmidhuber, 1997) without external memory. We pass the low-dimensional embeddings $z_{t=1} \ldots z_{t=T}$ directly to the LSTM, and generate a prediction $\hat{y}$ by passing the final hidden state through an output layer.

Next we consider two alternative external memory architectures: the *Neural Turing Machine* (NTM; Graves et al. (2014)) and *Metalearned Neural Memory* (MNM; Munkhdalai et al. (2019)). This comparison allows us to determine to what extent our results depend on the specific details of the ESBN's external memory, and, in particular, the separation between its two information-processing pathways vs. the mere presence of an external memory. Our NTM implementation consists of an LSTM controller (which takes image embeddings as input, and generates a prediction $\hat{y}$ as output) that interacts with an external memory using both content-based and location-based read/write mechanisms. Our MNM implementation employs the publicly available code from the original paper, modified so as to employ the same encoder architecture and TCN procedure as the other architectures that we test. Just as with the ESBN, we allow both of these architectures an extra time step to process the information retrieved from memory following the final input.

We also consider the *Relation Net* (RN; Santoro et al. (2017)), an architecture that has proven to be an effective approach for a wide range of relational reasoning tasks. In our implementation, we treat the low-dimensional image embeddings as individual 'objects' in the RN framework, using a

shared MLP to process all pair-wise combinations of these embeddings, summing the outputs from this MLP, and then passing them to another MLP that generates the prediction $\hat{y}$.

We also compare our model against the *Transformer* (Vaswani et al., 2017), an architecture originally developed in the domain of natural language processing, but has proven to be effective for a wide range of sequential data, and demonstrated a capacity for some degree of extrapolation (Saxton et al., 2019). After applying the transformer architecture to the sequence of image embeddings (allowing self-attention between these embeddings), we compute an average of the (transformed) embeddings, and then pass this to a small MLP that then generates the task output $\hat{y}$.

Finally, we consider the *PrediNet* (Shanahan et al., 2019). PrediNet was designed with the goal of being 'explicitly relational,' and has been shown to be effective at generalizing learned relations to novel entities. We apply the PrediNet's multi-head attention over the 1D temporal sequence of image embeddings (as opposed to applying attention over a 2D image, as in the original work), and then pass the output of the PrediNet module to a small MLP that generates $\hat{y}$.

## 4 RESULTS

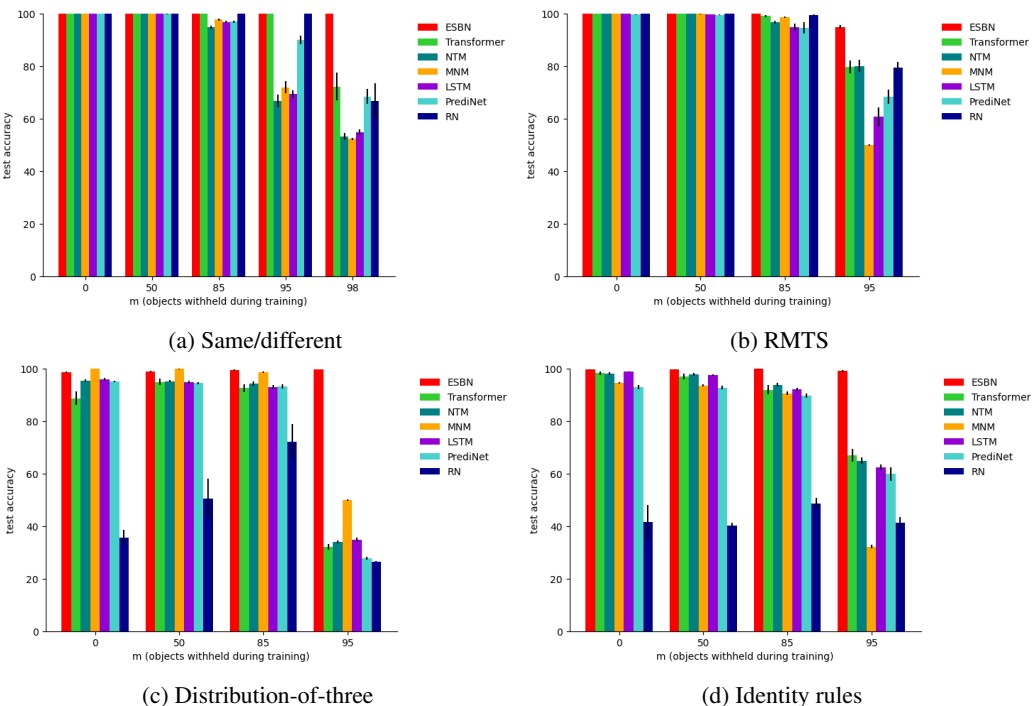

(a) Same/different

(b) RMTS

(c) Distribution-of-three

(d) Identity rules

Figure 3: Results for all four tasks with $m$ objects withheld (out of $n = 100$) during training. Results reflect test accuracy averaged over 10 trained networks ($\pm$ the standard error of the mean).

Figure 3 shows the generalization results for all four tasks. Our primary finding is that the ESBN displayed nearly perfect generalization ($\geq 95\%$) of the learned rule in all four tasks, even when trained on a very limited number of problems (just hundreds of problems, in the case of the most extreme generalization regimes) involving a limited number of entities (as few as just two entities, in the case of the same/different task), and tested on completely novel entities. Some of the alternative architectures that we evaluated showed a surprising capability to generalize to novel entities in some tasks as seen, for instance, in the generalization results for the Transformer and RN on the same/different and RMTS tasks (though we note that all architectures incorporate TCN, without which generalization is significantly worse, as shown in A.5.1). Nevertheless, none of these alternative architectures were able to generalize what they learned in the most extreme generalization regimes, whereas the ESBN performed comparably well across all regimes.

Notably, the RN performed very poorly on the distribution-of-three and identity rules tasks, even in the easiest regime ($m = 0$). We speculate that this results from the fact that the RN is biased toward pair-wise relations, whereas these tasks are both based on a ternary relation. It is possible to represent this ternary relation as a combination of pair-wise relations, but doing so requires a more complex strategy and therefore likely more training data. We include results in A.5.2 demonstrating that the RN is capable of successfully generalizing in this task (though not in the most extreme regimes) when trained on an order of magnitude more data ($10^5$ instead of $10^4$ examples). We also present results for the Temporal Relation Network (Zhou et al., 2018), an RN variant that incorporates ternary relations via subsampling, though we find that this doesn't help as much as increasing the amount of training data.

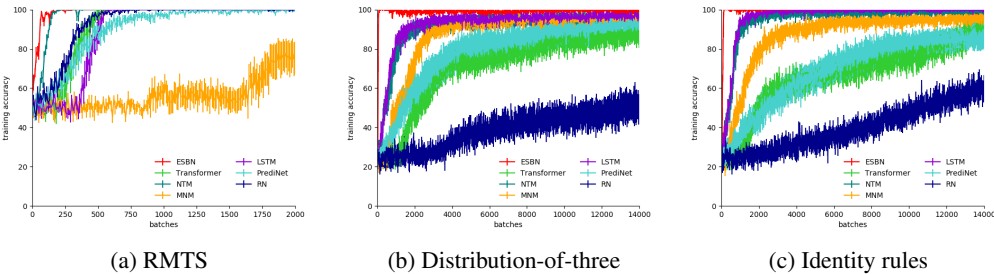

|          (a) RMTS          |   (b) Distribution-of-three   |      (c) Identity rules      |

Figure 4: Training accuracy time courses for all models on the $m = 0$ regime. Each time course reflects an average over 10 trained networks. Error bars reflect the standard error of the mean.

In addition to requiring a very small amount of training data and generalizing systematically to novel entities, the ESBN also requires very little training time. Figure 4 shows training accuracy time courses for the RMTS, distribution-of-three, and identity rules tasks for all models [3]. The ESBN converged to nearly perfect training accuracy within 100 to 200 training updates on all four tasks, whereas the other models required thousands, or even tens of thousands of training updates to reach convergence[4].

We also performed some experiments to better understand how the ESBN operates, and why it was so effective. First, we tested whether the systematic generalization exhibited by the ESBN was dependent on the use of convolutional layers in the encoder, which naturally confer a significant degree of generalization in tasks that involve shape recognition. We found that the ESBN generalized to novel entities comparably well when using either an MLP encoder or a random projection (see A.5.4 for details), suggesting that the ESBN is capable of generalizing learned rules to any arbitrary set of entities, regardless of how those entities are encoded. For comparison, we also performed the same experiments with the Transformer (the best performing alternative architecture on our tasks) and found that, by contrast, its performance was significantly impaired by the use of a random projection instead of a convolutional encoder.

Second, we performed an ablation experiment on the confidence value appended to retrieved memories. We found that ablation of these confidence values impaired the ESBN's performance in both the same/different and RMTS tasks, but not the distribution-of-three or identity rules tasks (see A.5.5 for details). A likely reason for this result is that the distribution-of-three and identity rules tasks only require retrieval of the best match from memory, whereas the same/different and RMTS tasks require a sense of how good of a match that memory is, which is exactly the information that the confidence value conveys. This dissociation mirrors the distinction sometimes made in cognitive psychology between recollection and familiarity (Yonelinas, 2001).

Third, we performed an analysis of the key representations learned by the controller. We hypothesized that the controller would learn to represent abstract variables in the keys that it writes to memory, and that these representations therefore shouldn't vary based on the values to which they are bound. This analysis revealed a high degree of overlap between the keys written during training

---

[3]We found that all models converged within a few hundred training updates on the relatively easy same/different discrimination task, so those time courses are omitted here, but shown in A.5.3.

[4]Note that all models eventually reached convergence, though this required that some of them be trained longer than is depicted in these figures, as detailed in A.4.

and test (involving entirely different entities), suggesting that this was indeed the case (see A.6 for details). This ability to arbitrarily bind values to variables, without affecting the representations of those variables, is a key property of symbol-processing systems, and is likely the basis of the strong systematic generalization exhibited by the ESBN.

## 5    RELATED WORK

There have been a number of proposals for augmenting neural networks with an external memory. An influential early line of work, Complementary Learning Systems (McClelland et al., 1995), proposed that neural systems benefit from having components that learn on different time scales, and argues that this combination allows neural networks both to learn general, abstract structure (using standard learning algorithms) and to rapidly encode arbitrary new items (using an external memory). In recent years, there have been a number of proposals for how to implement the latter efficiently, including Fast Weights (Ba et al., 2016a), the NTM and closely related Differentiable Neural Computer (Graves et al., 2016), and the Differentiable Neural Dictionary (DND; Pritzel et al. (2017)). Our external memory approach is most closely related to the DND, which also involves a two-column key/value memory. Variations on key/value memory have also been employed in other more recently proposed approaches, such as the Memory Recall Agent (Fortunato et al., 2019) and the Dual-Coding Episodic Memory (Hill et al., 2020), where it afforded various benefits in terms of generalization. One critical difference between our model and this previous work is that the ESBN's controller is forced to interact with perceptual inputs only indirectly through its memory, a design decision that we argue is crucial to its ability to systematically generalize what it learns.

It is worth noting that architectures such as Fast Weights and the NTM are, in principle, capable of implementing variable-binding, though it is a separate question whether such a strategy will result from learning in any particular task. Along these lines, a recent study from Chen et al. (2019) found that both of these architectures are capable of generalizing learned structure to novel entities when allowed a sufficiently dense sampling of the space of potential objects (the 'objects' in their study were randomly sampled 50-dimensional vectors). This contrasts with our findings, in which the NTM performed poorly when trained on far fewer samples from a much higher-dimensional space (in the $m = 95$ regime). This suggests that indirection and variable-binding, though possible in principle for architectures such as the NTM, do not emerge in practice when given only a limited amount of training experience, whereas this capacity is explicitly built into the ESBN.

At a high level, the idea of factoring a model into two distinct information processing streams, one that codes abstract task-relevant variables or roles and one that codes concrete entities, has been explored before. Kriete et al. (2013) proposed the PBWM Indirection model, in which one population of neurons acted as a pointer to another population of neurons by gating its activity, and showed that this model enabled a significant degree of generalization to novel role/filler bindings. Whittington et al. (2019) proposed the Tolman-Eichenbaum machine, a model that is capable of learning abstract relational structure (such as 2D spatial maps), and showed that this model captured a number of phenomena relating to grid cells and place cells. Russin et al. (2019) proposed Syntactic Attention, an architecture involving separate pathways for processing syntax vs. semantics, and showed that this approach was capable of a significant degree of compositional generalization on the challenging SCAN dataset. Relative to this previous work, our central contribution is the development of a simple model that can learn abstract rules directly from high-dimensional data (images), exploiting this same high-level idea to enable nearly perfect generalization of those rules to novel entities.

There has been extensive modeling work focusing on some of the tasks that we study. The recent development of two datasets modeled after Raven's Progressive Matrices, Procedurally Generated Matrices (Barrett et al., 2018), and RAVEN (Zhang et al., 2019), has spurred the development of models that are capable of solving RPM-like problems (Jahrens & Martinetz, 2020; Wu et al., 2020). However, these models typically require very large training sets (on the order of $10^6$ training examples), and largely fail to generalize outside of the specific conditions under which they are trained, whereas the ESBN exhibits the ability to learn rapidly and generalize out-of-distribution.

There have also been a number of models proposed to account for the human ability to rapidly learn identity rules (Alhama & Zuidema, 2019). Though some of these models achieved significant generalization of learned identity rules to novel entities, they did so mostly through the inclusion of

highly task-specific mechanisms. By contrast, our aim in the present work was to present a general approach that could be applied to a wider range of tasks.

Finally, there have been a number of recent proposals for so-called 'neurosymbolic' models, incorporating elements from both the neural network and symbolic modeling frameworks (Mao et al., 2019; Nye et al., 2020). Though we have emphasized the notion of 'emergent symbols' in the present work, we stress that this is quite distinct from neurosymbolic modeling efforts since we do not explicitly incorporate any symbolic machinery into the ESBN. Instead, our approach was to show how the functional equivalent of symbols can emerge in a neural network model with an appropriate architecture and binding mechanism.

# 6  DISCUSSION

## 6.1  LIMITATIONS AND FUTURE WORK

One open question is whether the strict division between the two information processing streams in the ESBN is necessary, and whether it limits the sorts of relations and rules that it can learn. In future work, it may be desirable to soften this division, for instance by encouraging it in a regularization term, rather than strictly enforcing it architecturally.

A second limitation is that the tasks we study are not as complex as other similar tasks that have recently been studied, such as the two recently proposed RPM-like benchmarks (Barrett et al., 2018; Zhang et al., 2019). In the present work, we intentionally stripped away some of this complexity in order to make progress on the issue of out-of-distribution generalization. Extending the ESBN to more complex tasks will likely require the incorporation of visual attention mechanisms to enable selective sequential processing of individual elements within a scene. There are many recently proposed approaches for doing this (Gregor et al., 2015; Locatello et al., 2020). In future work, we look forward to extending the ESBN in this manner and testing it on more complex tasks.

## 6.2  RELATION TO WORK IN NEUROSCIENCE

It is worth considering how the present work relates to pre-existing theories of how the brain might implement variable-binding. Classic proposals for variable-binding in neural systems emphasize dynamic binding of representations, either by computing the tensor product between those representations (Smolensky, 1990), or by establishing synchronous activation between two pools of units (Hummel & Holyoak, 1997). An alternative proposal is that variable-binding is accomplished via semi-permanent synaptic changes in the hippocampus, relying on the same mechanism that plays a central role in episodic memory (Cer & O'Reilly, 2006). This approach relies on contextual information and retrieval processes to prevent potential interference between conflicting memories, rather than explicit unbinding mechanisms. Our model is more in line with the latter account, since it does not possess an unbinding mechanism. As such, our model can be seen as part of a recent trend toward the reinterpretation of putatively working memory functions in terms of episodic memory (Beukers et al., 2020).

# 7  CONCLUSION

In this work, we have presented a model of abstract rule learning based on a novel architecture, the ESBN, and shown that this model is capable of rapidly learning abstract rules directly from images given only a small amount of training experience, and then successfully generalizing those rules to novel entities. Key to the model's performance is its separation into two streams that only interact through indirection, allowing the ESBN to learn tasks in a manner that is abstracted away from the specific entities involved, and resulting in the emergence of symbol-like representations. We believe that these results suggest that such a variable-binding capacity is an essential ingredient for achieving human-like abstraction and generalization, and hope that the ESBN will be a useful tool for doing so.

## ACKNOWLEDGMENTS

We would like to thank Zachary Dulberg, Steven Frankland, Randall O'Reilly, Alexander Petrov, and Simon Segert for their helpful feedback and discussions.

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

# A APPENDIX

## A.1 CODE AVAILABILITY

All code, including code for dataset generation, model implementation, training, and evaluation, is available on GitHub.

## A.2 DATASET GENERATION

In this section, we provide details on the dataset generation process for all tasks. In all of our simulations, a dataset was generated from scratch (according to the procedures described below) at the beginning of each training run, such that different runs involved different datasets, though the statistics were the same across these datasets. We did this to prevent the possibility that our results would reflect biases present in a particular dataset.

Table 1: Training and test set sizes for the same/different discrimination task.

|  | $m = 0$ | $m = 50$ | $m = 85$ | $m = 95$ | $m = 98$ |
|---|---|---|---|---|---|
| Training | 18,810 | 4,900 | 420 | 40 | 4 |
| Test | 990 | 4,900 | 10,000 | 10,000 | 10,000 |

### A.2.1 SAME/DIFFERENT DISCRIMINATION

Given $n = 100$ total images, with $m = 0$ withheld during training, there are $n^2 = 10^4$ possible same/different problems. To prevent the potential for networks to be biased by the fact that the overwhelming majority of these are 'different' problems, we created balanced datasets by including duplicates of the 'same' problems. Specifically, we randomly sampled (with replacement) $n(n-1)$ of the $n$ unique 'same' trials and combined them with the $n(n-1)$ unique 'different' trials, resulting in $2n(n-1) = 19,800$ total problems. We reserved 990 of these problems for test, yielding training sets including $18,810$ problems (ensuring that duplicates of the same problem did not appear in both the training and test sets).

We followed a similar procedure for the other regimes, generating balanced datasets by duplicating the 'same' problems when necessary. These datasets incorporated either all of the problems that resulted from this procedure (given the $n-m$ images available for training, or the $m$ images available for test), or $10,000$ problems, whichever was smaller. The exact size of each of these datasets is shown in Table 1.

### A.2.2 RMTS

Table 2: Training and test set sizes for the RMTS task.

|  | $m = 0$ | $m = 50$ | $m = 85$ | $m = 95$ |
|---|---|---|---|---|
| Training | 10,000 | 10,000 | 10,000 | 480 |
| Test | 10,000 | 10,000 | 10,000 | 10,000 |

For the RMTS task, we constructed balanced training and test sets ensuring that there were an equal number of problems with a 'same' vs. 'different' source pair. These datasets contained either $10,000$ problems, or the minimum number of problems possible in a given regime, whichever was smaller (Table 2). For most regimes, $10,000$ problems constitutes a tiny fraction of the full space of possible problems (ranging from $10^9$ for the $m = 0$ regime to $10^5$ for the training set in the $m = 85$ regime), and thus there was no need to duplicate problems to achieve balanced datasets. For the $m = 95$ regime, there are only $480$ possible training problems, which happen to include the same number of 'same' and 'different' trial types.

### A.2.3 DISTRIBUTION-OF-THREE

Table 3: Training and test set sizes for the distribution-of-three task.

|  | $m = 0$ | $m = 50$ | $m = 85$ | $m = 95$ |
|---|---|---|---|---|
| Training | 10,000 | 10,000 | 10,000 | 360 |
| Test | 10,000 | 10,000 | 10,000 | 10,000 |

For the distribution-of-three task, we generated problems by randomly selecting three of the available images in a given regime (either $n - m$ images during training, or $m$ images during test), and then randomly sampling two permutations of those images for the two rows (allowing the possibility

that the same permutation appears in both rows) of the $2 \times 3$ matrix. We then randomly selected a fourth image to appear with the other three as possible answers, and randomly permuted these four answer choices. When taking into account the identity of this fourth image, and the permutation of the answer choices, the number of unique distribution-of-three problems in the $m = 0$ regime is on the order of $10^{10}$.

For most regimes, we randomly created training and test sets consisting of $10,000$ randomly sampled problems. For the $m = 95$ regime, the training set consisted of 360 problems (the total number of unique problems possible in this regime when not considering the identity and order of the answer choices, which were randomly selected).

### A.2.4 IDENTITY RULES

Table 4: Training and test set sizes for the identity rules task.

|          | $m = 0$ | $m = 50$ | $m = 85$ | $m = 95$ |
|----------|---------|----------|----------|----------|
| Training | 10,000  | 10,000   | 10,000   | 8,640    |
| Test     | 10,000  | 10,000   | 10,000   | 10,000   |

For the identity rules task, we constructed datasets with an approximately balanced (through uniform random sampling) number of ABA, ABB, and AAA problems. These datasets consisted of either $10,000$ problems, or the minimum number of possible problems in a given regime, whichever was smaller (Table 4). For the training set in the $m = 95$ regime, datasets consisting of $8,640$ problems were constructed from the $7,200$ possible unique problems in this regime, by duplicating the AAA problems to match the number of ABA/ABB problems. For all other datasets, $10,000$ problems constituted a small fraction of the total number of possible problems (ranging from $10^9$ for the $m = 0$ regime to $10^6$ for the training set in the $m = 85$ regime), and no duplication was necessary to achieve balanced problem types.

### A.3 IMPLEMENTATION DETAILS FOR ALL MODELS

#### A.3.1 ENCODER

We used the same feedforward encoder architecture to process each of the images in a sequence $x_{t=1} \ldots x_{t=T}$, generating low-dimensional embeddings $z_{t=1} \ldots z_{t=T}$ that were then passed to the core sequential component of each model (either the ESBN or one of the alternative architectures)[5]. This encoder consisted of three convolutional layers, each with 32 channels, a $4 \times 4$ kernel, and a stride of 2, followed by two fully-connected layers with 256 units and 128 units respectively. All layers used ReLU nonlinearities. All weights were initialized using a Kaiming normal distribution (He et al., 2015), and all biases were initialized to 0.

#### A.3.2 TASK OUTPUT LAYER

All models had an output layer for generating $\hat{y}$. The number of units and the nonlinearity depended on the task. For the same/different and RMTS tasks, the output layer had 1 unit and a sigmoid nonlinearity (producing a number between 0 and 1 to code for 'same' vs. 'different', or pair 1 vs. pair 2). For the distribution-of-three and identity rules tasks, the output layer had 4 units and a softmax nonlinearity (to select 1 of the 4 answer choices). The weights of the output layer were initialized using an Xavier normal distribution (Glorot & Bengio, 2010), and the biases were initialized to 0.

#### A.3.3 ESBN

The details of the ESBN's operations are given in Algorithm 1. The LSTM controller had 1 layer with 512 units, and employed the standard tanh nonlinearities and sigmoidal gates. The controller

---

[5]The encoder architecture was the same for all models, but the parameters of the encoder were learned end-to-end from scratch on each training run.

also had output layers for $\boldsymbol{k}_w$ (256 units with a ReLU nonlinearity), $g$ (1 unit with a sigmoid nonlinearity), and $\hat{\boldsymbol{y}}$. The input to the controller at each time step was $\boldsymbol{k}_r$, the key retrieved from memory at the previous time step (along with the associated confidence value, $c_k$). At the beginning of each sequence, $\boldsymbol{k}_r$ and the controller's hidden state $\boldsymbol{h}$ were initialized to $0$.

After processing a full sequence, the ESBN was allowed an additional time step for the controller to process the retrieved key associated with the final input. After this additional time step, the final hidden state of the LSTM was passed through the task output layer to generate the prediction $\hat{\boldsymbol{y}}$. We note that it is also possible to retrieve values from memory (from $\boldsymbol{M}_v$) using a similar procedure and then decode these values to make predictions in image space (Sinha et al., 2020), but in the present work we focus only on the classification component of the model.

The input weights for the LSTM controller were initialized using an Xavier normal distribution with a gain value of $5/3$. The weights for the LSTM's gates, as well as the weights for the output layer for the gate $g$, were initialized with an Xavier normal distribution (with a gain of $1$). The weights for the output layer that produced $\boldsymbol{k}_w$ were initialized using a Kaiming normal distribution. All biases were initialized to $0$. The parameters $\gamma$ and $\beta$ were initialized to $1$ and $0$ respectively.

### A.3.4   LSTM

The LSTM architecture had 1 layer with 512 units. Image embeddings were passed to the LSTM as a sequence, after which $\hat{\boldsymbol{y}}$ was generated through a task output layer. The LSTM's hidden state was initialized to $0$ at the beginning of each sequence. The LSTM's weights were initialized using the same scheme as the LSTM controller in the ESBN (using an Xavier normal distribution, with a gain of $5/3$ for the input weights and $1$ for the gates), and biases were initialized to $0$.

### A.3.5   NTM

The NTM had an LSTM controller (1 layer with 512 units). The LSTM's hidden state was initialized to $0$ at the beginning of each sequence, and the LSTM's parameters were initialized in the same way as the LSTM architecture and the controller for the ESBN. The NTM had one write head and one read head. The read head had the following output layers:

1. read key: 256 units, tanh nonlinearity, weights initialized using an Xavier normal distribution with a gain of $5/3$.
2. key strength: 1 unit, softplus nonlinearity, weights initialized using a Kaiming normal distribution.
3. interpolation gate: 1 unit, sigmoid nonlinearity, weights initialized using an Xavier normal distribution with a gain of $1$.
4. shift weights: 3 units (corresponding to the allowable shifts $-1$, $0$, and $1$), softmax nonlinearity, weights initialized using an Xavier normal distribution with a gain of $1$.

The write head had the following output layers:

1. erase vector: 256 units, sigmoid nonlinearity, weights initialized using an Xavier normal distribution with a gain of $1$.
2. add vector: 256 units, tanh nonlinearity, weights initialized using an Xavier normal distribution with a gain of $5/3$.
3. write key: 256 units, tanh nonlinearity, weights initialized using an Xavier normal distribution with a gain of $5/3$.
4. key strength: 1 unit, softplus nonlinearity, weights initialized using a Kaiming normal distribution.
5. interpolation gate: 1 unit, sigmoid nonlinearity, weights initialized using an Xavier normal distribution with a gain of $1$.
6. shift weights: 3 units (corresponding to the allowable shifts $-1$, $0$, and $1$), softmax nonlinearity, weights initialized using an Xavier normal distribution with a gain of $1$.

All biases for these output layers were initialized to $0$. The NTM used these outputs to interact with its external memory, employing all of the location- and content-based mechanisms described in the original work (Graves et al., 2014). Cosine similarity was used as a similarity measure for the content-based mechanisms. The memory matrix had 10 rows of size 256. The initial state of the

memory at the beginning of each sequence was learned. The learned initial state was initialized (at the beginning of training) using an Xavier normal distribution.

The input to the LSTM controller at each time step consisted of the image embedding corresponding to that time step and the read vector from the previous time step. At the beginning of each sequence, the read vector, read weights, and write weights were initialized to 0. Just as with the ESBN, the NTM was allowed an additional time step to process the read vector retrieved from memory after observing the final image embedding, after which $\hat{y}$ was generated through an output layer from the LSTM controller.

### A.3.6  MNM

We implemented the MNM using publicly available code released with the original publication (Munkhdalai et al., 2019). Specifically, we used the version of MNM that employs a learned local update ('MNM-p' in the original paper). Before passing the images in our tasks to the MNM model, we applied the same encoder and TCN procedure used for the other architectures that we tested. Other than this modification, the original implementation, including all architectural hyperparameters, was unmodified.

### A.3.7  RN

The RN implementation consisted of two MLPs. The first MLP (used to process all pair-wise combinations of image embeddings) had a hidden layer of size $512$ and an output layer of size $256$. The outputs from the first MLP were summed, and then passed to the second MLP, which had a hidden layer of size $256$ and an output layer for generating $\hat{y}$. All layers (except the output layer) used ReLU nonlinearities. All weights were initialized using a Kaiming normal distribution (except the output layer, which was initialized according to the description in A.3.2), and all biases were initialized to $0$. Before passing the image embeddings to the first MLP, they were appended with a tag (an integer from $0$ to $T - 1$) indicating their position in the input sequence.

### A.3.8  TRN

The TRN employs two key design decisions intended to prevent the combinatorial explosion that would naturally result from the inclusion of n-ary relations:

1. Only considering temporally ordered, non-redundant sets (whereas the original RN considers all possible pairs of objects, including both permutations of the same pair, and the pair of each object with itself).
2. Subsampling from these sets.

We found that it was computationally feasible to implement a TRN with ternary relations in our tasks by only using (1), without the need to subsample. Thus, our implementation considers all temporally ordered, non-redundant sets of two and three.

Each pair of image embeddings was processed by an MLP with a hidden layer of size $512$ and an output layer of size $256$. The outputs of this MLP for all pairs were then summed, and processed by an additional fully-connected layer with $256$ units, yielding a single vector representing all pairs.

Each set of three image embeddings was processed by a separate MLP with the same hyperparameters (hidden layer of $512$ units, output layer of $256$ units). The outputs of this MLP for all sets of three were then summed, and processed by a separate fully-connected layer with $256$ units, yielding a single vector representing all sets of three.

These two vectors, representing all pairs and sets of three, were then summed and passed to an additional fully-connected layer with $256$ units, and then to the output layer to generate $\hat{y}$. All layers (except for the output layer) used ReLU nonlinearities. All weights in these layers were initialized using a Kaiming normal distribution, and all biases were initialized to $0$.

Just as with the RN, we append the image embeddings with a tag indicating their position in the sequence before passing them to the first MLP in the TRN.

### A.3.9 TRANSFORMER

The Transformer implementation consisted of a single Transformer encoder layer. We also experimented with 2- and 3-layer Transformers but these did not generalize as well as the 1-layer Transformer in the tasks that we studied. Positional encoding (as described by Vaswani et al. (2017)) was applied to the sequence of image embeddings, which were then passed to the Transformer layer. The self-attention layer had 8 heads. The MLP had a single hidden layer with 512 units, and used ReLU nonlinearities. Residual connections and layer normalization (Ba et al., 2016b) were used following both the self-attention layer and the MLP. The self-attention weights (for generating the keys, queries, and values) were initialized using an Xavier normal distribution. The MLP weights were initialized using a Kaiming normal distribution.

After applying the Transformer layer, the (transformed) embeddings were averaged and passed to an output MLP. The output MLP had a single hidden layer with 256 units, and an output layer for generating $\hat{y}$. The hidden layer used ReLU nonlinearities, and the weights were initialized using a Kaiming normal distribution. All biases were initialized to 0.

### A.3.10 PREDINET

The PrediNet implementation was as close as possible to the model described in the original work (Shanahan et al., 2019), except that the multi-head attention was applied over the 1D temporal sequence of image embeddings, rather than over a 2D feature map (since there was no spatial component to the tasks that we studied). Before being passed to the PrediNet module, the image embeddings were appended with a tag (an integer from 0 to $T-1$) indicating their temporal position. The PrediNet module used keys of size 16, 32 heads, and 16 relations. All weights in the PrediNet module were initialized using an Xavier normal distribution.

The output of all PrediNet heads was concatenated and passed to an output MLP. This MLP had a single hidden layer with 8 units, and an output layer for generating $\hat{y}$. The hidden layer used ReLU nonlinearities, and the weights were initialized using a Kaiming normal distribution. All biases were initialized to 0.

### A.4 TRAINING DETAILS

Table 5: Learning rates for all models trained without TCN.

|  | Same/different | RMTS | Distribution-of-three | Identity rules |
|---|---|---|---|---|
| ESBN | $5e^-5$ | $5e^-5$ | $5e^-5$ | $5e^-5$ |
| Transformer | $5e^-4$ | $5e^-4$ | $5e^-4$ | $5e^-4$ |
| NTM | $5e^-4$ | $5e^-4$ | $5e^-4$ | $5e^-4$ |
| MNM | $5e^-4$ | $5e^-4$ | $5e^-4$ | $5e^-4$ |
| LSTM | $5e^-4$ | $5e^-4$ | $5e^-4$ | $5e^-4$ |
| PrediNet | $5e^-4$ | $5e^-4$ | $5e^-5$ | $5e^-5$ |
| RN | $5e^-4$ | $5e^-5$ | $5e^-4$ | $5e^-4$ |

All models were trained with a batch size of 32 using the ADAM optimizer (Kingma & Ba, 2014). The learning rate for all models trained with TCN was $5e^-4$. Some of the models failed to converge when trained without TCN, requiring a smaller learning rate of $5e^-5$. The learning rates used for all models when trained without TCN are shown in Table 5.

Because different generalization regimes (different values for $m$) involved different training set sizes, and therefore involved fewer training updates per epoch, the number of training epochs required to reliably achieve convergence varied based on the regime. The default number of training epochs for all tasks and regimes is shown in Table 6.

Some models required additional training on some tasks to reach convergence. The PrediNet and the RN required longer training on the distribution-of-three task (Table 7), and the PrediNet, RN, and Transformer required longer training on the identity rules task (Table 8).

Table 6: Default number of training epochs for all tasks and regimes.

|  | $m = 0$ | $m = 50$ | $m = 85$ | $m = 95$ | $m = 98$ |
|---|---|---|---|---|---|
| Same/different | 50 | 50 | 50 | 100 | 100 |
| RMTS | 50 | 50 | 50 | 200 | – |
| Distribution-of-three | 50 | 50 | 50 | 150 | – |
| Identity rules | 50 | 50 | 50 | 50 | – |

Table 7: Number of training epochs for the PrediNet and RN on the distribution-of-three task.

|  | $m = 0$ | $m = 50$ | $m = 85$ | $m = 95$ |
|---|---|---|---|---|
| PrediNet | 100 | 100 | 100 | 150 |
| RN | 150 | 150 | 150 | 800 |

Table 8: Number of training epochs for the PrediNet, RN, and Transformer on the identity rules task.

| $m = 0$ | $m = 50$ | $m = 85$ | $m = 95$ |
|---|---|---|---|
| 100 | 100 | 100 | 150 |

When training the RN on larger datasets for the distribution-of-three and identity rules tasks, the same learning rate and number of training epochs as used when training on smaller datasets was sufficient to reach convergence.

## A.5 SUPPLEMENTARY RESULTS

### A.5.1 RESULTS WITH AND WITHOUT TCN

Tables 9 - 12 show the results for all models trained both with and without TCN. With the exception of the PrediNet on the same/different task, every model benefited on every task from the incorporation of TCN, in many cases substantially. Results for models trained with TCN (indicated by '+ TCN') correspond to the results presented in Figure 3 (except for the results of the PrediNet on the same/different task, for which the version of the model trained without TCN is plotted in Figure 3).

We note that, even with a lower learning rate of $5e^{-}5$, some models failed to converge without TCN, such as the ESBN on the same/different task, or the RN on the RMTS task. It is possible that some of these models might have performed better if we had optimized them further by training for longer or trying different learning rates, but we opted not to do that since TCN was so effective across all of the models and tasks that we studied.

Table 9: Results for same/different task. Results reflect test accuracy averaged over 10 trained networks ($\pm$ the standard error of the mean).

| | $m = 0$ | $m = 50$ | $m = 85$ | $m = 95$ | $m = 98$ |
|---|---|---|---|---|---|
| ESBN + TCN | $100.0 \pm 0.0$ | $100.0 \pm 0.0$ | $100.0 \pm 0.0$ | $100.0 \pm 0.0$ | $100.0 \pm 0.0$ |
| ESBN | $50.0 \pm 0.02$ | $50.0 \pm 0.0$ | $50.1 \pm 0.1$ | $49.8 \pm 0.2$ | $50.1 \pm 0.1$ |
| Transformer + TCN | $100.0 \pm 0.0$ | $100.0 \pm 0.0$ | $100.0 \pm 0.0$ | $100.0 \pm 0.0$ | $72.3 \pm 5.2$ |
| Transformer | $100.0 \pm 0.0$ | $99.9 \pm 0.02$ | $95.4 \pm 0.6$ | $73.7 \pm 1.8$ | $56.1 \pm 1.3$ |
| NTM + TCN | $100.0 \pm 0.0$ | $99.99 \pm 0.0$ | $94.9 \pm 0.6$ | $66.7 \pm 2.5$ | $53.3 \pm 1.4$ |
| NTM | $99.0 \pm 0.9$ | $98.6 \pm 0.3$ | $84.9 \pm 2.4$ | $57.0 \pm 2.2$ | $52.5 \pm 0.9$ |
| MNM + TCN | $100.0 \pm 0.0$ | $99.95 \pm 0.03$ | $97.8 \pm 0.4$ | $72.0 \pm 2.4$ | $52.3 \pm 0.5$ |
| MNM | $98.9 \pm 0.1$ | $95.1 \pm 1.8$ | $88.6 \pm 1.1$ | $59.1 \pm 1.6$ | $51.7 \pm 0.7$ |
| LSTM + TCN | $100.0 \pm 0.0$ | $99.97 \pm 0.01$ | $96.9 \pm 0.3$ | $69.4 \pm 1.5$ | $54.8 \pm 1.1$ |
| LSTM | $88.2 \pm 3.2$ | $97.0 \pm 0.5$ | $85.5 \pm 2.4$ | $61.8 \pm 1.7$ | $56.5 \pm 1.6$ |
| PrediNet + TCN | $100.0 \pm 0.0$ | $99.7 \pm 0.1$ | $96.0 \pm 1.3$ | $67.2 \pm 2.9$ | $61.6 \pm 2.3$ |
| PrediNet | $100.0 \pm 0.0$ | $99.9 \pm 0.03$ | $97.0 \pm 0.4$ | $90.0 \pm 1.6$ | $68.5 \pm 2.8$ |
| RN + TCN | $100.0 \pm 0.0$ | $100.0 \pm 0.0$ | $100.0 \pm 0.0$ | $99.9 \pm 0.04$ | $66.8 \pm 6.6$ |
| RN | $99.98 \pm 0.02$ | $98.5 \pm 0.4$ | $53.2 \pm 1.4$ | $50.5 \pm 0.2$ | $52.3 \pm 0.7$ |

Table 10: Results for relational match-to-sample task. Results reflect test accuracy averaged over 10 trained networks ($\pm$ the standard error of the mean).

| | $m = 0$ | $m = 50$ | $m = 85$ | $m = 95$ |
|---|---|---|---|---|
| ESBN + TCN | $100.0 \pm 0.0$ | $100.0 \pm 0.0$ | $100.0 \pm 0.0$ | $95.0 \pm 0.7$ |
| ESBN | $86.4 \pm 6.1$ | $69.4 \pm 6.5$ | $50.0 \pm 0.1$ | $51.0 \pm 0.5$ |
| Transformer | $100.0 \pm 0.0$ | $99.98 \pm 0.01$ | $99.1 \pm 0.4$ | $79.8 \pm 2.5$ |
| Transformer | $99.4 \pm 0.1$ | $96.8 \pm 0.7$ | $86.4 \pm 1.9$ | $49.9 \pm 0.2$ |
| NTM + TCN | $100.0 \pm 0.0$ | $99.97 \pm 0.01$ | $96.8 \pm 0.5$ | $80.1 \pm 2.3$ |
| NTM | $99.5 \pm 0.1$ | $92.5 \pm 4.7$ | $81.2 \pm 1.5$ | $50.1 \pm 0.2$ |
| MNM + TCN | $99.99 \pm 0.0$ | $99.9 \pm 0.03$ | $98.7 \pm 0.3$ | $50.0 \pm 0.2$ |
| MNM | $74.6 \pm 7.6$ | $63.6 \pm 5.7$ | $78.3 \pm 3.7$ | $50.0 \pm 0.2$ |
| LSTM + TCN | $99.99 \pm 0.0$ | $99.8 \pm 0.03$ | $94.9 \pm 1.3$ | $60.7 \pm 3.7$ |
| LSTM | $99.1 \pm 0.3$ | $90.2 \pm 2.0$ | $80.9 \pm 1.1$ | $50.2 \pm 0.1$ |
| PrediNet + TCN | $99.7 \pm 0.1$ | $99.6 \pm 0.1$ | $94.6 \pm 2.2$ | $68.4 \pm 2.7$ |
| PrediNet | $54.9 \pm 4.7$ | $50.1 \pm 0.2$ | $65.9 \pm 3.7$ | $49.7 \pm 0.2$ |
| RN + TCN | $100.0 \pm 0.0$ | $99.99 \pm 0.0$ | $99.5 \pm 0.3$ | $79.6 \pm 2.1$ |
| RN | $50.1 \pm 0.2$ | $49.9 \pm 0.2$ | $50.2 \pm 0.2$ | $50.0 \pm 0.1$ |

Table 11: Results for distribution-of-three task. Results reflect test accuracy averaged over 10 trained networks ($\pm$ the standard error of the mean).

|                   | $m = 0$          | $m = 50$        | $m = 85$        | $m = 95$        |
| ----------------- | ---------------- | --------------- | --------------- | --------------- |
| ESBN + TCN        | $98.7 \pm 0.4$   | $99.0 \pm 0.3$  | $99.5 \pm 0.2$  | $99.7 \pm 0.1$  |
| ESBN              | $99.98 \pm 0.0$  | $97.4 \pm 0.2$  | $92.4 \pm 1.1$  | $62.0 \pm 4.0$  |
| Transformer + TCN | $88.7 \pm 2.6$   | $95.0 \pm 1.2$  | $92.7 \pm 1.5$  | $32.1 \pm 1.0$  |
| Transformer       | $62.1 \pm 3.3$   | $68.6 \pm 3.6$  | $72.6 \pm 4.4$  | $28.0 \pm 0.8$  |
| NTM + TCN         | $95.5 \pm 0.4$   | $95.2 \pm 0.4$  | $94.3 \pm 0.8$  | $34.0 \pm 0.5$  |
| NTM               | $92.9 \pm 0.5$   | $87.1 \pm 1.4$  | $78.2 \pm 1.4$  | $26.7 \pm 0.3$  |
| MNM + TCN         | $94.7 \pm 0.3$   | $93.6 \pm 0.4$  | $90.6 \pm 0.7$  | $32.2 \pm 0.6$  |
| MNM               | $58.5 \pm 8.9$   | $68.7 \pm 6.2$  | $48.4 \pm 5.5$  | $25.6 \pm 0.3$  |
| LSTM + TCN        | $96.0 \pm 0.6$   | $94.8 \pm 0.5$  | $92.9 \pm 0.8$  | $34.8 \pm 0.8$  |
| LSTM              | $91.3 \pm 0.6$   | $85.3 \pm 1.5$  | $71.6 \pm 4.3$  | $27.5 \pm 0.3$  |
| PrediNet + TCN    | $95.2 \pm 0.3$   | $94.6 \pm 0.4$  | $93.3 \pm 0.9$  | $27.8 \pm 0.5$  |
| PrediNet          | $75.1 \pm 3.0$   | $65.7 \pm 7.4$  | $78.0 \pm 6.0$  | $25.7 \pm 0.1$  |
| RN + TCN          | $35.6 \pm 3.0$   | $50.6 \pm 7.6$  | $72.2 \pm 6.8$  | $26.5 \pm 0.3$  |
| RN                | $25.1 \pm 0.1$   | $24.9 \pm 0.1$  | $25.7 \pm 0.3$  | $25.2 \pm 0.1$  |

Table 12: Results for identity rules task. Results reflect test accuracy averaged over 10 trained networks ($\pm$ the standard error of the mean).

|                   | $m = 0$          | $m = 50$        | $m = 85$        | $m = 95$        |
| ----------------- | ---------------- | --------------- | --------------- | --------------- |
| ESBN + TCN        | $99.6 \pm 0.2$   | $99.6 \pm 0.1$  | $99.9 \pm 0.04$ | $99.2 \pm 0.4$  |
| ESBN              | $100.0 \pm 0.0$  | $99.4 \pm 0.1$  | $97.8 \pm 0.2$  | $95.2 \pm 0.4$  |
| Transformer + TCN | $98.3 \pm 0.7$   | $97.1 \pm 1.0$  | $92.0 \pm 1.7$  | $67.1 \pm 2.4$  |
| Transformer       | $75.5 \pm 4.1$   | $71.6 \pm 5.1$  | $85.4 \pm 4.6$  | $38.6 \pm 2.2$  |
| NTM + TCN         | $98.2 \pm 0.6$   | $97.8 \pm 0.5$  | $93.9 \pm 0.6$  | $64.9 \pm 1.2$  |
| NTM               | $94.6 \pm 0.3$   | $90.1 \pm 0.8$  | $82.2 \pm 1.2$  | $25.0 \pm 0.1$  |
| MNM + TCN         | $95.2 \pm 0.4$   | $93.8 \pm 0.4$  | $90.8 \pm 0.5$  | $61.5 \pm 1.5$  |
| MNM               | $70.9 \pm 10.2$  | $69.5 \pm 9.7$  | $49.8 \pm 8.4$  | $24.9 \pm 0.2$  |
| LSTM + TCN        | $98.9 \pm 0.1$   | $97.7 \pm 0.3$  | $92.1 \pm 0.7$  | $62.5 \pm 1.1$  |
| LSTM              | $93.8 \pm 0.5$   | $89.3 \pm 0.6$  | $73.7 \pm 5.7$  | $24.8 \pm 0.1$  |
| PrediNet + TCN    | $93.0 \pm 0.8$   | $92.8 \pm 0.7$  | $89.8 \pm 0.8$  | $59.9 \pm 2.6$  |
| PrediNet          | $40.8 \pm 0.4$   | $40.5 \pm 1.9$  | $40.3 \pm 2.2$  | $32.2 \pm 0.6$  |
| RN + TCN          | $41.5 \pm 6.7$   | $40.2 \pm 1.0$  | $48.7 \pm 2.0$  | $41.4 \pm 2.0$  |
| RN                | $41.1 \pm 7.2$   | $37.3 \pm 3.4$  | $31.6 \pm 2.8$  | $25.4 \pm 0.4$  |

A.5.2 PERFORMANCE OF RN ON TERNARY RELATIONS

Table 13 shows the results for the RN (w/ TCN) on the distribution-of-three and identity rules tasks when trained on larger training sets ($10^5$ instead of $10^4$ training examples). These results show that with more training data, the RN, which is biased toward processing pair-wise relations, is able to learn these tasks (which are based on ternary relations) in a manner that enables some degree of generalization. Note that these results do not include the $m = 95$ regime, because there are not enough images in that regime to create larger training sets than were originally used.

Table 13: Results for the RN on the distribution-of-three and identity rules tasks when trained on a larger training set. Results reflect test accuracy averaged over 10 trained networks ($\pm$ the standard error of the mean).

|  | $m = 0$ | $m = 50$ | $m = 85$ |
|---|---|---|---|
| Distribution-of-three | $84.5 \pm 10.0$ | $84.6 \pm 9.9$ | $72.4 \pm 9.7$ |
| Identity rules | $89.2 \pm 5.0$ | $99.7 \pm 0.1$ | $86.6 \pm 4.1$ |

We also tested the TRN, which incorporates ternary relations through subsampling, on these tasks (with the standard training set size of $10^4$ training examples). Table 14 shows the results. This yielded a slight improvement over the RN (when trained on $10^4$ training examples), though not as much of an improvement as resulted from training the RN with a larger training set. This result may seem surprising given that the TRN explicitly incorporates ternary relations. We note two possible explanations for this result:

1. The systematic comparison of every pair of objects, including permutations and comparisons of each object with itself, allows the RN to take advantage of a very powerful form of data augmentation, enforcing a certain degree of systematicity in the relations that it learns. By only considering temporally ordered and non-redundant sets, the TRN is not able to take advantage of this to the same extent, and therefore might not learn relations that generalize as well.
2. The distribution-of-three and identity rules tasks both involve not only ternary sets, but the higher-order comparison of multiple pairs of ternary sets (the first row vs. the combination of the second row with each candidate answer). One could presumably engineer a solution to this problem within the RN framework, but we take it as a strength of the ESBN that no such special engineering is necessary in this case.

Table 14: Results for the TRN on the distribution-of-three and identity rules tasks. Results reflect test accuracy averaged over 10 trained networks ($\pm$ the standard error of the mean).

|  | $m = 0$ | $m = 50$ | $m = 85$ | $m = 95$ |
|---|---|---|---|---|
| Distribution-of-three | $60.2 \pm 5.4$ | $77.5 \pm 5.7$ | $88.7 \pm 0.8$ | $27.8 \pm 0.5$ |
| Identity rules | $40.3 \pm 2.0$ | $43.6 \pm 2.0$ | $52.8 \pm 2.7$ | $44.9 \pm 1.0$ |

A.5.3 TRAINING TIME COURSES FOR SAME/DIFFERENT TASK

Figure 5 shows the training time courses for all models on the same/different task. Unlike the other three tasks we studied (for which training time courses are shown in Figure 4), all models were able to learn this task within a few hundred training updates (though all models except the ESBN failed to generalize in the most extreme regime).

A.5.4 ALTERNATIVE ENCODER ARCHITECTURES

In order to determine whether the systematic generalization exhibited by the ESBN depended to some extent on the convolutional layers in its encoder, we performed experiments with two alternative encoder architectures: a multilayer perceptron (MLP) encoder, and a random projection.

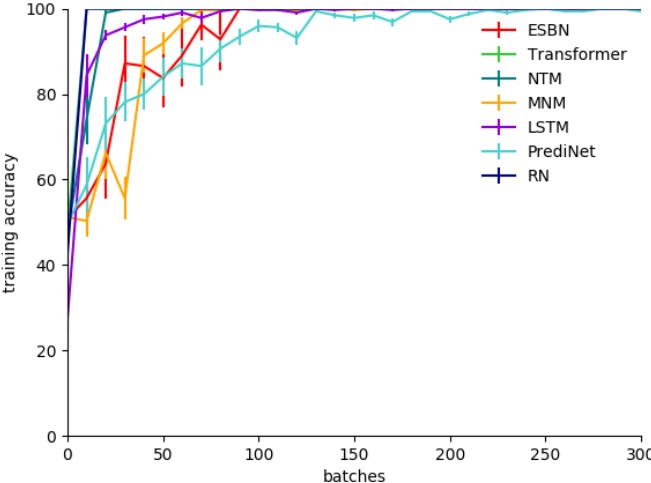

Figure 5: Training accuracy time courses on $m = 0$ regime of the same/different task. Each time course reflects an average over 10 trained networks. Error bars reflect the standard error of the mean.

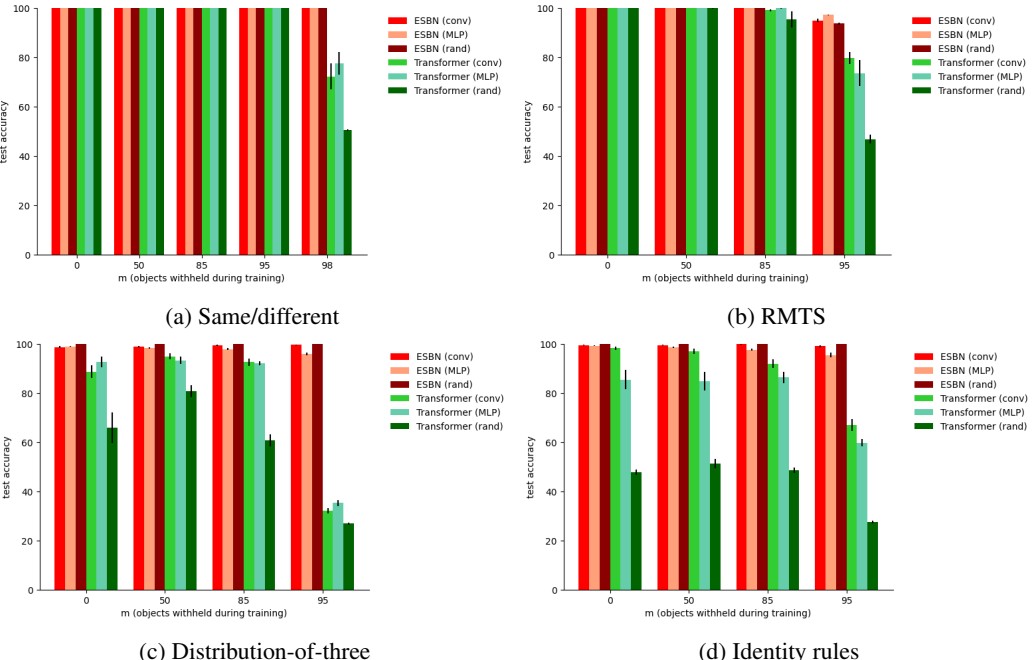

(a) Same/different

(b) RMTS

(c) Distribution-of-three

(d) Identity rules

Figure 6: Results for all four tasks with convolutional (conv), multilayer perceptron (MLP), or random (rand) encoders. Results reflect test accuracy averaged over 10 trained networks ($\pm$ the standard error of the mean).

The MLP encoder consisted of 3 fully-connected layers, with $512$, $256$, and $128$ units, each of which used ReLU nonlinearities. All weights were initialized using a Kaiming normal distribution, and all biases were set to $0$.

The random projection encoder involved only a single, untrained, fully-connected layer that projected from the flattened image to $128$ units, followed by a ReLU nonlinearity. Weights were sampled from a Kaiming normal distribution, and biases were set to $0$.

Figure 6 and Tables 15 - 18 show the results for these experiments, along with the original version of the model (with a convolutional encoder) for comparison. To enable a fair comparison with the original model, all experiments employed TCN. The results show that the ESBN performed comparably well with all three of the encoder architectures. This was confirmed by performing paired t-tests on the average test accuracy in each task/generalization condition (each combination of task and value of $m$) for the MLP vs. convolutional encoder ($t = -1.7$, $p = 0.1$) and for the random vs. convolutional encoder ($t = 1.6$, $p = 0.13$).

For comparison, we also performed experiments with these alternative encoders in the Transformer architecture. These experiments revealed that, in contrast with the ESBN, the Transformer's performance was significantly impaired by the use of a random vs. convolutional encoder ($t = -4.0$, $p = 0.001$), though it appeared to perform comparably well with an MLP vs. convolutional encoder ($t = -1.6$, $p = 0.14$).

Table 15: Results for same/different task with convolutional (conv), multilayer perceptron (MLP), or random (rand) encoders. Results reflect test accuracy averaged over 10 trained networks ($\pm$ the standard error of the mean).

|  | $m = 0$ | $m = 50$ | $m = 85$ | $m = 95$ | $m = 98$ |
|---|---|---|---|---|---|
| ESBN (conv) | $100.0 \pm 0.0$ | $100.0 \pm 0.0$ | $100.0 \pm 0.0$ | $100.0 \pm 0.0$ | $100.0 \pm 0.0$ |
| ESBN (MLP) | $100.0 \pm 0.0$ | $100.0 \pm 0.0$ | $100.0 \pm 0.0$ | $100.0 \pm 0.0$ | $100.0 \pm 0.0$ |
| ESBN (rand) | $100.0 \pm 0.0$ | $100.0 \pm 0.0$ | $100.0 \pm 0.0$ | $100.0 \pm 0.0$ | $100.0 \pm 0.0$ |
| Transformer (conv) | $100.0 \pm 0.0$ | $100.0 \pm 0.0$ | $100.0 \pm 0.0$ | $100.0 \pm 0.0$ | $72.3 \pm 5.2$ |
| Transformer (MLP) | $100.0 \pm 0.0$ | $100.0 \pm 0.0$ | $100.0 \pm 0.0$ | $100.0 \pm 0.0$ | $77.6 \pm 4.6$ |
| Transformer (rand) | $100.0 \pm 0.0$ | $100.0 \pm 0.0$ | $100.0 \pm 0.0$ | $100.0 \pm 0.0$ | $50.6 \pm 0.3$ |

Table 16: Results for relational match-to-sample task with convolutional (conv), multilayer perceptron (MLP), or random (rand) encoders. Results reflect test accuracy averaged over 10 trained networks ($\pm$ the standard error of the mean).

|  | $m = 0$ | $m = 50$ | $m = 85$ | $m = 95$ |
|---|---|---|---|---|
| ESBN (conv) | $100.0 \pm 0.0$ | $100.0 \pm 0.0$ | $100.0 \pm 0.0$ | $95.0 \pm 0.7$ |
| ESBN (MLP) | $100.0 \pm 0.0$ | $100.0 \pm 0.0$ | $100.0 \pm 0.0$ | $97.2 \pm 0.2$ |
| ESBN (rand) | $100.0 \pm 0.0$ | $100.0 \pm 0.0$ | $100.0 \pm 0.0$ | $93.8 \pm 0.4$ |
| Transformer (conv) | $100.0 \pm 0.0$ | $99.98 \pm 0.01$ | $99.1 \pm 0.4$ | $79.8 \pm 2.5$ |
| Transformer (MLP) | $100.0 \pm 0.0$ | $100.0 \pm 0.0$ | $99.9 \pm 0.1$ | $73.6 \pm 5.3$ |
| Transformer (rand) | $99.99 \pm 0.01$ | $99.9 \pm 0.04$ | $95.4 \pm 3.2$ | $46.8 \pm 1.7$ |

Table 17: Results for distribution-of-three task with convolutional (conv), multilayer perceptron (MLP), or random (rand) encoders. Results reflect test accuracy averaged over 10 trained networks ($\pm$ the standard error of the mean).

|  | $m = 0$ | $m = 50$ | $m = 85$ | $m = 95$ |
|---|---|---|---|---|
| ESBN (conv) | $98.7 \pm 0.4$ | $99.0 \pm 0.3$ | $99.5 \pm 0.2$ | $99.7 \pm 0.1$ |
| ESBN (MLP) | $99.0 \pm 0.1$ | $98.4 \pm 0.3$ | $98.0 \pm 0.3$ | $95.9 \pm 0.5$ |
| ESBN (rand) | $100.0 \pm 0.0$ | $100.0 \pm 0.0$ | $100.0 \pm 0.0$ | $100.0 \pm 0.0$ |
| Transformer (conv) | $88.7 \pm 2.6$ | $95.0 \pm 1.2$ | $92.7 \pm 1.5$ | $32.1 \pm 1.0$ |
| Transformer (MLP) | $92.7 \pm 2.1$ | $93.3 \pm 1.5$ | $92.1 \pm 0.8$ | $35.3 \pm 1.2$ |
| Transformer (rand) | $66.0 \pm 6.2$ | $80.8 \pm 2.5$ | $60.9 \pm 2.4$ | $26.9 \pm 0.4$ |

Table 18: Results for identity rules task with convolutional (conv), multilayer perceptron (MLP), or random (rand) encoders. Results reflect test accuracy averaged over 10 trained networks ($\pm$ the standard error of the mean).

|  | $m = 0$ | $m = 50$ | $m = 85$ | $m = 95$ |
|---|---|---|---|---|
| ESBN (conv.) | $99.6 \pm 0.2$ | $99.6 \pm 0.1$ | $99.9 \pm 0.04$ | $99.2 \pm 0.4$ |
| ESBN (MLP) | $99.3 \pm 0.2$ | $98.6 \pm 0.3$ | $97.7 \pm 0.4$ | $95.5 \pm 1.0$ |
| ESBN (random) | $100.0 \pm 0.0$ | $100.0 \pm 0.0$ | $100.0 \pm 0.0$ | $100.0 \pm 0.0$ |
| Transformer (conv.) | $98.3 \pm 0.7$ | $97.1 \pm 1.0$ | $92.0 \pm 1.7$ | $67.1 \pm 2.4$ |
| Transformer (MLP) | $85.5 \pm 4.0$ | $84.8 \pm 3.8$ | $86.4 \pm 2.3$ | $59.8 \pm 1.5$ |
| Transformer (random) | $47.8 \pm 1.1$ | $51.4 \pm 1.9$ | $48.6 \pm 1.1$ | $27.5 \pm 0.6$ |

### A.5.5 Confidence ablation experiment

In order to determine the importance of the confidence values appended to retrieved memories, we tested a version of the ESBN without these confidence values. These results are shown in Table 19 and Figure 7. The ablation of confidence values prevented the ESBN from being able to perform the same/different task at all, and resulted in much slower training on the RMTS task. By contrast, ablation of confidence values did not affect performance, either in terms of generalization or training time, for the distribution-of-three or identity rules tasks. This can be explained by the fact that these tasks only require the retrieval of the best match from memory, whereas the same/different and RMTS tasks require the model to know how good of a match the best match is, which is precisely the information conveyed by confidence values.

Table 19: Results for the confidence ablation experiment. Results reflect test accuracy averaged over 10 trained networks ($\pm$ the standard error of the mean).

|  | $m = 0$ | $m = 50$ | $m = 85$ | $m = 95$ | $m = 98$ |
|---|---|---|---|---|---|
| Same/different | $50.0 \pm 0.02$ | $50.0 \pm 0.0$ | $50.0 \pm 0.05$ | $49.8 \pm 0.1$ | $50.0 \pm 0.1$ |
| RMTS | $99.95 \pm 0.01$ | $99.9 \pm 0.02$ | $99.9 \pm 0.02$ | $96.0 \pm 0.6$ | $-$ |
| Distribution-of-three | $99.2 \pm 0.2$ | $99.0 \pm 0.3$ | $99.5 \pm 0.3$ | $99.8 \pm 0.1$ | $-$ |
| Identity rules | $99.6 \pm 0.1$ | $99.6 \pm 0.2$ | $99.8 \pm 0.1$ | $99.2 \pm 0.2$ | $-$ |

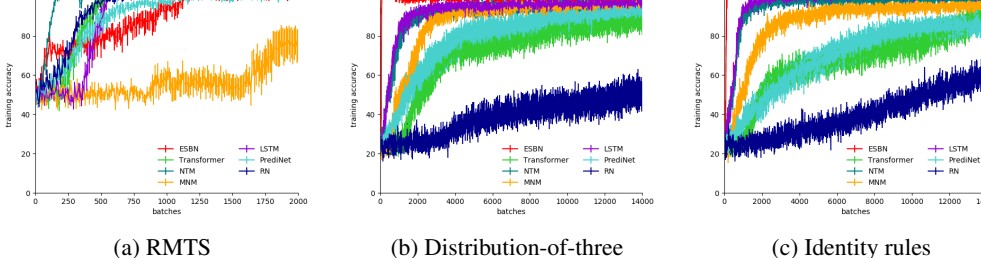

(a) RMTS  (b) Distribution-of-three  (c) Identity rules

Figure 7: Training accuracy time courses for the ESBN model without confidence values on the $m = 0$ regime, shown with the time courses for all other models for comparison. Each time course reflects an average over 10 trained networks. Error bars reflect the standard error of the mean.

It is also worth noting one potential alternative to an explicit, inbuilt confidence value. In our implementation, the ESBN's memory is empty at the beginning of each sequence that it processes. However, when multiple entries are present in memory, as will generally be the case in realistic, temporally extended settings, the presentation of a previously unseen item will result in the retrieval of a mixture of (weakly matched) memories. This mixed representation can therefore serve as a

reliable cue for the degree to which the current percept matches a stored memory, obviating the need for an explicit confidence value. To demonstrate this, we implemented a version of the ESBN that begins each sequence with a single, learned key/value entry stored in memory (initialized to $0$ at the beginning of training). Table 20 shows that this approach allows the ESBN to learn and perfectly generalize on the same/different task. Figure 8 shows that this approach allows the ESBN to retain the short training time of the original model on the RMTS task.

Table 20: Results on the same/different task for the ESBN model with a learned default memory instead of confidence values. Results reflect test accuracy averaged over 10 trained networks ($\pm$ the standard error of the mean).

| $m = 0$ | $m = 50$ | $m = 85$ | $m = 95$ | $m = 98$ |
|---|---|---|---|---|
| $100.0 \pm 0.0$ | $100.0 \pm 0.0$ | $100.0 \pm 0.0$ | $100.0 \pm 0.0$ | $100.0 \pm 0.0$ |

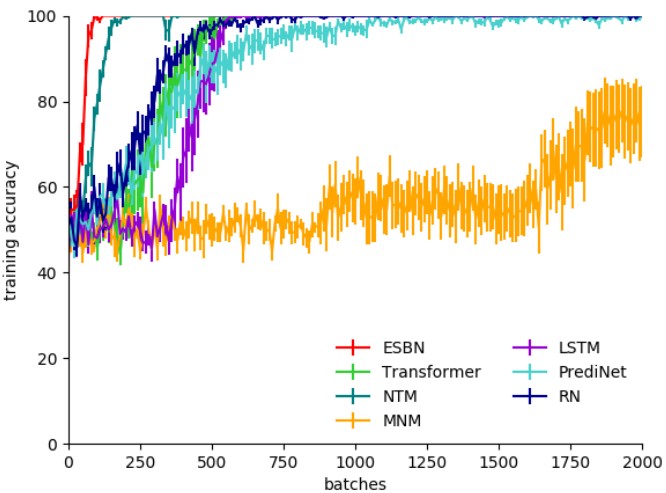

Figure 8: Training accuracy time courses on $m = 0$ regime of the RMTS task for the ESBN model with a learned default memory instead of confidence values. Each time course reflects an average over 10 trained networks. Error bars reflect the standard error of the mean.

A.6    ANALYSIS OF LEARNED REPRESENTATIONS

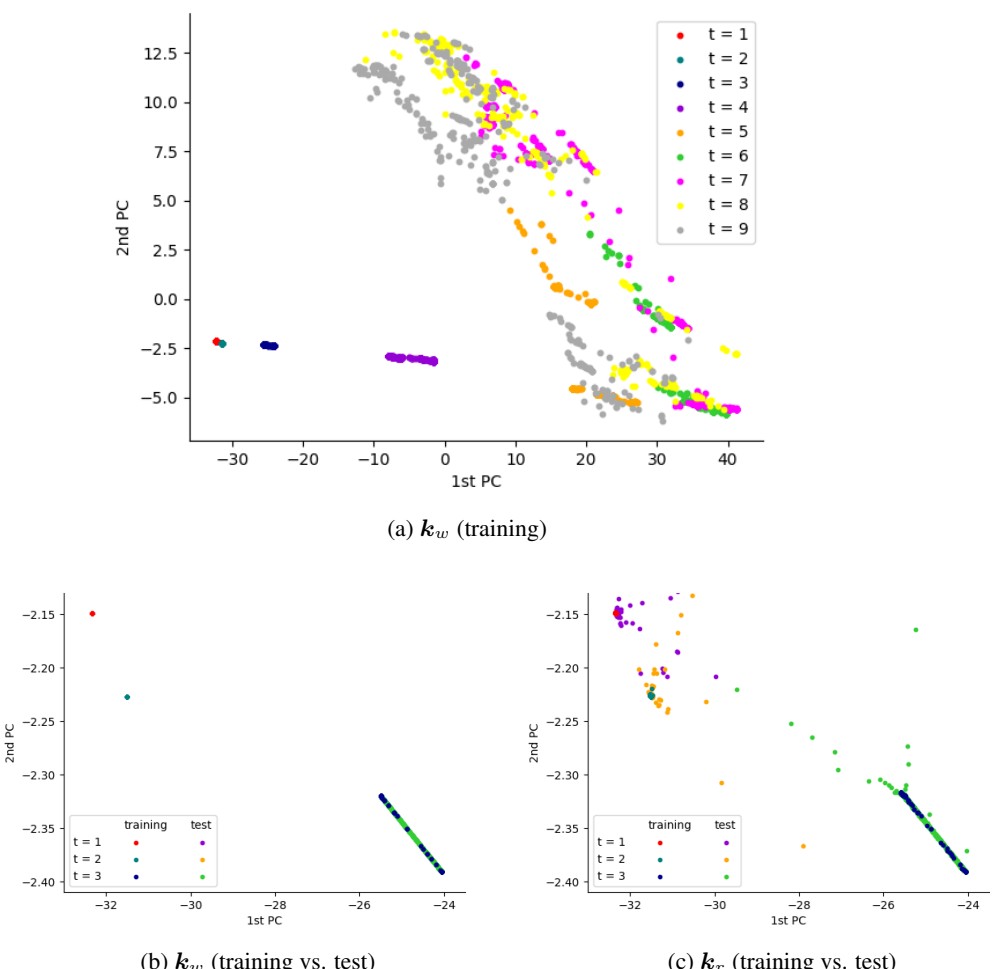

(a) $\boldsymbol{k}_w$ (training)

(b) $\boldsymbol{k}_w$ (training vs. test)          (c) $\boldsymbol{k}_r$ (training vs. test)

Figure 9: Representations learned by ESBN (projected along first two principal components). (a) Keys written to memory during time steps 1-9 (training set). (b) Keys written to memory during time steps 1-3 (training set vs. test set). (c) Keys retrieved from memory following second appearance of objects that first appeared during time steps 1-3 (training set vs. test set).

To better understand how the ESBN works, we performed an analysis of the representations that it learned on the distribution-of-three task. Specifically, we performed an analysis of a network trained on the most difficult generalization regime ($m = 95$), by performing principal component analysis (PCA) on all key vectors written to and retrieved from memory for both the training and test sets, and visualizing these vectors along the first two principal components.

First, we looked at the keys that were written to memory ($\boldsymbol{k}_w$). We found that the keys for the first three time steps were tightly clustered, whereas the keys for the subsequent time steps (4-9) were more diffuse (Figures 9a and 9b). This makes sense because, in the distribution-of-three task, the ESBN only needs to be able to reliably retrieve what it wrote during the first three time steps (when the objects in the first row were presented). For time steps 4-9, the only important consideration is that the keys written to memory not overlap with those written during the first three time steps, which also appears to be the case.

Second, we compared the keys written to memory for the first three time steps in the training vs. test sets (Figure 9b). This revealed that, for a given time step, the keys written to memory in the

training vs. test sets were remarkably similar (so much so that they are completely overlapping for time steps 1 and 2).

Third, we looked at the keys that were retrieved from memory following the second appearance of the objects that appeared on time steps 1-3. We found that 1) these closely matched the distribution of keys written to memory during time steps 1-3, and 2) these were highly overlapping for the training vs. test sets (Figure 9c).

Taken together, these results help to explain why the ESBN was so successful in this generalization regime, despite the very small degree of overlap between the distribution of training and test images. Because the ESBN's controller was relatively isolated from the part of the model that deals with image embeddings, it was able to learn to encode abstract symbol-like representations (such as 'first image', 'second image', and 'third image'), that did not depend on the identity of the images. Then, when queried with an image, was able to successfully retrieve the image's corresponding abstract encoding, even when that image was quite different than those observed during training. That is, the model learned representations to use as keys that could be used for binding and indirection in the same way that symbols are used in traditional computational architectures.

## A.7 UNICODE CHARACTERS

Figure 10 shows all 100 images that were used to construct the abstract rule learning tasks.

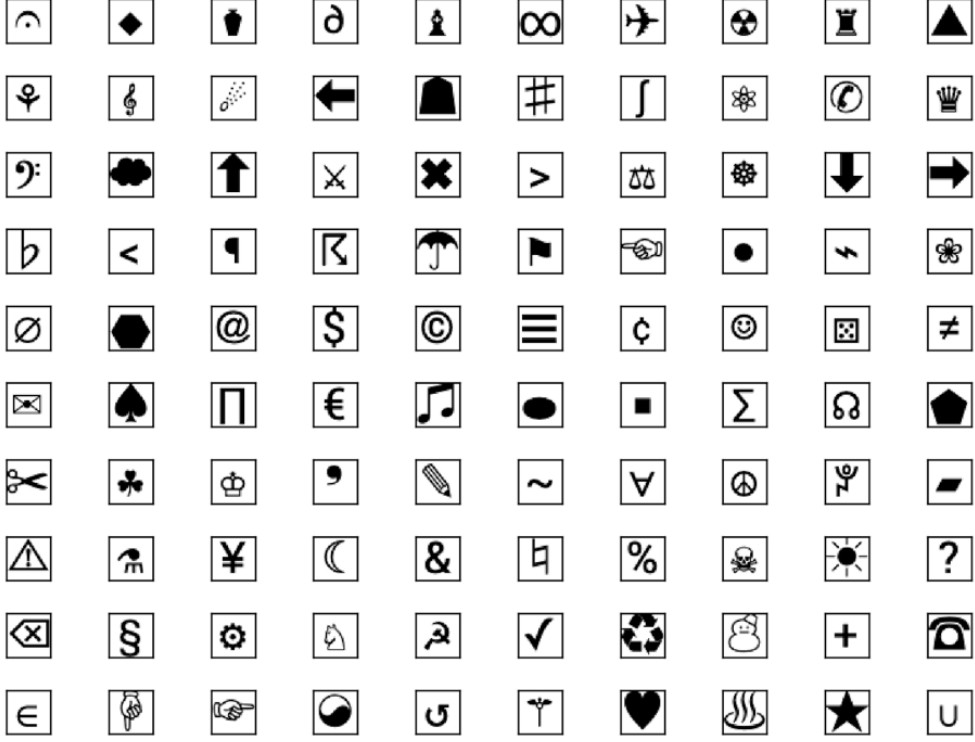

Figure 10

