# OpenReview forum: "Emergent Symbols through Binding in External Memory"
_ICLR.cc/2021/Conference — ICLR 2021 Spotlight_

### Official Review · AnonReviewer4 · 2020-10-27
**Review of "Emergent Symbols through Binding in External Memory"**

**Rating:** 6
**Confidence:** 4

**Review:**

Summary:

The authors present an architecture capable learning symbolic representations and rules over those symbols.  Also presented is a set of  tasks involving manipulation of symbolic rules over a set of symbols where held symbol sets may be used to measure generalization performance on the task set.  The goal of this approach is learn from high dimensional inputs useful symbolic representations which are difficult to generalize across from pixels alone.

The  architecture described in this paper is named the emergent symbol binding network (ESBN) which is composed of a two column external memory for indirection and two information processing streams, one for representing embeddings of concrete variables and another that is recurrent and trained to operate over task relevant variables.  The architecture uses Temporal Context Normalization across sequences to strengthen the models sensitivity to out-of-distribution samples.  This architecture is then trained and evaluated across a set of tasks involving symbolic entities and rules over which the ESBN must learn to generalize and is compared to other memory based architectures with a capacity for relational reasoning.


Strength & Weaknesses:

The architecture divides the  memory into symbolic keys and pixel encoded values and allows gradients to flow between them (symbols -> encodings) via a similarity weighting function.  As keys are generated from an LSTM they should also encode the sequence history and learn to store the correct information over the sequence to perform the symbolic manipulation required to determine the correct answers to the tasks set out.

The architecture to me seems not entirely novel to me.  There have been other approaches using key-value based episodic memories, for example the architecture in Fortunato et al. (2019) "Generalization of reinforcement learners with working and episodic memory".  The particular way that symbols are learned is interesting but I don't think necessarily exclusive to this model. I also have some concerns the model overfit the task.  It may have been useful to consider other the more complex tasks involving symbolic reasoning such as CLEVR or bAbI.  The tasks presented seem mainly to be toy tasks.

The addition of the Temporal Context Normalization for improving generalization over out-of-distribution samples is a nice.  It'd be interesting to see an ablation to quantify the effect of this, for instance, how does it compare with Batch normal or when simply removed for example.

I'm surprised that the Relation Net failed at this task - were positional encodings included in the representation?  same with the transformer - this is crucial when order matters in the input sequence.

The RN performing poorly on the distribution-of-three and identity-rules tasks is surprising and might be due to 1) the relationships are limited to two entities or 2) the exclusion of sequence order info.  As such, I believe the results may not fully be reflecting the Relation Nets capacity on such a problem as I'd expect them to be able to solve these problems  e.g. The RN could easily be extended to consider all ternary relations and is mainly limited by the task definition.

More equations would have been helpful.  Possibly consider including Algorithm 1 in the main paper as it is the only description of how the model works.


Recommendation:

While I believe that this is a very interesting problem domain I think that the solution provided would have been more compelling if the scope of the problem was more ambitious and also I believe that the baselines could have been stronger.

---

> ### Author Response · Authors · 2020-11-23
> **Reply to Reviewer 4**
>
> We would like to thank the reviewer for the insightful and thorough comments on our work. The reviewer notes that there have been other approaches that rely on a key/value external memory, such as [1] (which we now cite and discuss in paragraph 1 of section 5). We certainly do not claim to have originated the idea to separate memory into keys and values, and have made our best effort to emphasize in our discussion of related work that this is not what is novel about our proposed approach. The key novelty of the proposed ESBN architecture, as we have now attempted to clarify in our discussion, is that the controller is forced to interact with incoming perceptual data only indirectly through its memory, allowing it to learn task-specific procedures that are abstract with respect to those inputs. We are not aware of any previous work that has employed this approach and showed that it is capable of learning abstract rules in a manner that generalizes systematically to novel entities.
>
> The reviewer suggested that it would be interesting to test how the ESBN performs on tasks such as CLEVR and bAbi, and expressed some concern about the relative simplicity of the tasks that we use in the present work. We agree that it would be very interesting to evaluate the ESBN on these tasks in future work, and in particular to evaluate whether it could afford a similar benefit in terms of out-of-distribution generalization on those tasks. However, in the present work, we intentionally designed tasks that were as simple as possible, while still capturing the key property of out-of-distribution generalization that we hoped to test, thus allowing us to focus on that property as a causal factor. Though the tasks we used are relatively simple when compared with other benchmarks such as CLEVR, generalizing effectively in these tasks was extremely challenging even for the more competitive baselines that we evaluated, which suggests that the tasks do successfully capture this key property. That  the relative simplicity of these tasks made it easier to understand how the ESBN operates is evidenced by our analysis of the representations that it learns.
>
> The reviewer noted that it would be nice to see how the various models we tested perform without temporal context normalization (TCN). We note that these results are included in the appendix (section A.4.1), where we show that in every case except for one (the PrediNet on the same/different task), TCN significantly improves generalization. We did not conduct a more thorough comparison of TCN with other normalization techniques, in part because the benefits of TCN were not the central focus of our work, and in part because the authors of the paper in which TCN was originally described [2] already performed such a comparison and found that TCN indeed outperformed other normalization techniques in terms of out-of-distribution generalization.

---

> > ### Author Response · Authors · 2020-11-23
> > **(continued)**
> >
> > The reviewer expressed some reservation about the fact that the relation net (RN) performed poorly on the distribution-of-three and identity rules tasks. We were also surprised by this result, and performed some followup experiments in order to better understand why the RN struggled on these tasks. First, we would like to note that we did indeed include positional encodings in our implementations, both for the RN (we appended a tag indicating the position in the temporal sequence, as described in [3]) and for the Transformer (using the scheme based on sine and cosine encodings described in [4]). Second, the reviewer suggests that the performance of the RN on these tasks might be improved by the inclusion of ternary comparisons. We believe that the emphasis on pair-wise relations is indeed why the RN struggled on these tasks, but it is not so straightforward to incorporate ternary relations into the RN architecture, at least as it is originally formulated. This is because the original RN formulation involves the consideration of every possible pair of objects in a given problem, even permutations of the same pair, and each object paired with itself. This rapidly leads to a combinatorial explosion when attempting to incorporate higher-order relations. Even the inclusion of ternary relations alone would require the evaluation of > 800 binary and ternary sets per problem for the relatively simple tasks that we study, at least when employing the original definition of RNs. One alternative is to only consider temporally ordered and non-redundant sets, as is the case for the Temporal Relation Network (TRN) [5]. We found that it was feasible to incorporate ternary comparisons given this simplification, and applied the TRN to our tasks. This improved performance on these tasks to some extent, but the performance was still not as good as the other architectures. We discuss some potential reasons why this modified version of the RN was not more successful on these tasks in section A.4.2. Additionally, we trained a version of the standard (pair-wise) RN on an order of magnitude more training data, and found that this improved performance on these tasks even more than the inclusion of ternary relations, though, as with the other architectures we evaluated, this performance still failed to generalize well in the more extreme generalization regimes. In sum, we believe we have correctly implemented the RN, and have made a significant effort to test alternative approaches and to evaluate the model as fairly as possible.
> >
> > The reviewer suggested that the paper would be easier to understand with more equations, and with the algorithm moved to the main body of the manuscript. We agreed, and moved the algorithm to section 3.2, alongside the figure and textual description of the ESBN. We thank the reviewer for this suggestion.
> >
> > Finally, the reviewer notes that the baselines we evaluated could have been stronger. We appreciate this point, and in the revised manuscript have added two baselines: 1) the TRN, as discussed above, and presented in section A.4.2 of the revised manuscript, and 2) metalearned neural memory (MNM) [6], which performed about as well as the neural turing machine, as shown in section 4 of the revised manuscript. We believe that these, together with the ones presented in the original manuscript, constitute a fair and informative comparison of our model against relevant alternatives.
> >
> > [1] Fortunato, M., Tan, M., Faulkner, R., Hansen, S., Badia, A. P., Buttimore, G., ... & Blundell, C. (2019). Generalization of reinforcement learners with working and episodic memory. In Advances in Neural Information Processing Systems (pp. 12469-12478).
> >
> > [2] Webb, T. W., Dulberg, Z., Frankland, S. M., Petrov, A. A., O'Reilly, R. C., & Cohen, J. D. (2020). Learning representations that support extrapolation. arXiv preprint arXiv:2007.05059.
> >
> > [3] Santoro, A., Raposo, D., Barrett, D. G., Malinowski, M., Pascanu, R., Battaglia, P., & Lillicrap, T. (2017). A simple neural network module for relational reasoning. In Advances in neural information processing systems (pp. 4967-4976).
> >
> > [4] Vaswani, A., Shazeer, N., Parmar, N., Uszkoreit, J., Jones, L., Gomez, A. N., ... & Polosukhin, I. (2017). Attention is all you need. In Advances in neural information processing systems (pp. 5998-6008).
> >
> > [5] Bolei Zhou, Alex Andonian, Aude Oliva, and Antonio Torralba. "Temporal relational reasoning in videos". In Proceedings of the European Conference on Computer Vision (ECCV), pages 803–818, 2018.
> >
> > [6] Munkhdalai, T., Sordoni, A., Wang, T., & Trischler, A. (2019). Metalearned neural memory. In Advances in Neural Information Processing Systems (pp. 13331-13342).

---

> > > ### Comment · AnonReviewer4 · 2020-11-24
> > > **Nice clarifications, Tasks and Results still borderline**
> > >
> > > Many thanks for addressing my concerns above.  It's very useful to know more details around the RN implementation and the point made regarding scaling to ternary inputs.  Also it is helpful to see a few more baselines.
> > >
> > > I believe that the authors have addressed some of the bigger issues in the paper around baselines and I am happy to raise my score to a five, however, I'm still left wondering what type of information is represented in the keys and how this might generalize into more more complex domains where symbolic information may be encoded in subtler ways, in the presence of greater noise in the task distribution.  It seems that modelling how symbolic mappings relate to representations is key to the power of the proposed model and given the results and the tasks I'm still not completely convinced but, the work looks promising.  I'm still on the fence, but if other reviewers have stronger opinions about this work I'd be fine to see it accepted.

---

> > > > ### Author Response · Authors · 2020-11-25
> > > > **Note on analyses of learned key representations**
> > > >
> > > > Thank you for the timely reply. To address the specific concern about what type of information is represented in the keys, we would just like to note that we included an analysis of this for the distribution-of-three task, as summarized in paragraph 6 of section 4 and described in detail of section A.5 of the revised manuscript. Specifically, we found that the learned keys formed tight clusters corresponding to the abstract roles played by the images (i.e. whether the images appeared in the first, second, or third cell of the first row), and, most importantly, that these learned representations were highly overlapping for the training vs. test sets, despite the fact that entirely different images were used. The reason this latter result is important is because it strongly suggests that the learned keys can be arbitrarily bound to novel entities, without affecting the key representations, which is an essential property of symbol-processing systems, and we believe, a major reason why the ESBN generalizes so well. We do agree that it is an interesting question what these learned symbol-like representations might look like in more complex tasks, and we look forward to addressing that question in future work.

---

### Official Review · AnonReviewer2 · 2020-10-27
**Nicely put together piece of work, but a couple more experiments might be needed**

**Rating:** 7
**Confidence:** 4

**Review:**

This work proposes a neural network architecture comprising a two-stream memory structure: one block is populated with visual representations, and the other is populated by hidden state vectors from a controller. The interaction between these two blocks is suggested to be akin to a symbolic-like mechanism for indirection. Reasoning that indirection is a central component for solving problems in the abstract, the authors go on to show that, indeed, such a network can solve abstract reasoning problems out-of-distribution.

A crucial aspect of the proposed network is that the memory keys are not at all conditioned on the current visual input. This means that the model needs to learn how to produce representations that will allow it to solve tasks in an abstract sense. This is a clever maneuver, because it solves the "out of distribution" problem by not allowing the information used to solve a task to ever become out of distribution: the keys, which condition the model's output, are always produced by the LSTM, and the LSTM is never conditioned on anything that is different at test time from what it sees during training time. The difficult task of handling out of distribution data, then, reduces to the vision encoder being able to simply acknowledge that one image is different from another, and it must do so using images it has never seen before. Such a mechanism is in contrast to fully connected networks that need to produce representations from out-of-distribution images *and* reason with those representations.

That being said, the out-of-distribution demands imposed on the visual encoder aren't as extreme as it would seem. This is because, for the current tasks, what is out-of-distribution is an object's *shape*. Convolutional encoders are predisposed through their inductive biases to detect shape differences. I would be quite interested to see how different encoders fair in this task, in this exact same two-memory network. For example, a simple random projection of the image, or an MLP would be interesting. I would hypothesize that the random projection method would work if images of the same type are identical, but would fail if images of the same type have subtle variation, as do, for example, a dataset of hand-written characters. Thinking about this further, it might be critical to also test the convolutional encoder on a dataset wherein shape types have variation between instances, because even the convolutional encoder might break down when "matching up" various instances of a type that are out-of-distribrution (whereas, in the work here, each representation it produces for a character type will necessarily be identical, even if the characters are out of distribution because each image instance is exactly the same).

The reason I emphasize these latter points about the encoder is because the work is pitched as providing a mechanism for allowing a certain type of symbolic-like reasoning, which in theory, should be insensitive to the types of encoders. If indirection is truly the causal mechanism at play, then the representations that do the "referring" should be completely insensitive to the representations that are "referred". But, we need evidence to demonstrate that this is the case.  Otherwise, we cannot claim that abstract variable binding occurs.

Altogether, the work is wonderfully put together. The experiments are simple, yet crisp, and they do the job. The work is well motivated, and well written, and there is ample background details to understand the models and experiments. Background work is adequately represented, and results are explained. My only reservations are in regards to the sensitivity of the encoder, and the sensitivity to the task design (particularly, the choice of using identical images per character type) to the overall performance.

---

> ### Author Response · Authors · 2020-11-23
> **Reply to Reviewer 2**
>
> We would like to thank the reviewer for the constructive feedback and lucid summary of our work. In particular, we appreciate the helpful suggestions for followup experiments. We took the suggestion and tested whether the ESBN was sensitive to the use of other types of encoders. These results are summarized in paragraph 4 of section 4, and described in detail in section A.4.4 of the revised manuscript. We found that the ESBN performed comparably well when using an MLP encoder or random projection instead of a convolutional encoder. By contrast, we found that the Transformer (on which we focused because it was the best performing alternative model on our tasks) was significantly impaired by the use of a random projection as an encoder. As we discuss in the revised manuscript, these results suggest that 1) the ESBN really is performing something like arbitrary variable-binding and 2) this is in contrast to the performance of the Transformer, which depends to some extent on the choice of encoder.
>
> The reviewer also suggested that we perform a version of our experiments that employ subtle variations between instances of each class e.g. handwritten characters. We agree that this would be a very interesting experiment to perform, and intend to do so in future work.

---

### Official Review · AnonReviewer3 · 2020-10-29
**An interesting work on improving out-of-distribution generalization**

**Rating:** 7
**Confidence:** 4

**Review:**

This work introduces a set of tasks that require a simple symbolic reasoning and a new model - Emergent Symbol Binding Network (ESBN) to solve them. The tasks are designed such that it tests the model’s extrapolation ability. Specifically, they introduce novel input symbols that are unseen during training to evaluate the model. They show that previous memory-augmented neural networks fail on the tasks whereas their approach with an external memory that supports a specific variable-binding achieves an excellent performance.

The main components of ESBN are an input encoder, a key-value memory and a LSTM controller.  The encoder is a CNN that takes an input symbol ((i.e. grayscale image) and outputs a value vector for the memory. The LSTM controller then produces a key vector by taking in a memory entry retrieved by using an attention mechanism. This retrieved memory entry consists of two parts: a key vector corresponding to a value that is selected by the attention and a confidence score. They calculate the attention and confidence scores based on the similarity between the current value and the other value vectors in the memory. The confidence score seems to be an important feature because it tells the LSTM that whether the new input symbol already existed in the memory or not.  Therefore, by design, the LSTM controller never sees the actual input symbols and it only sees their corresponding key vectors, which are in turn generated by the LSTM itself. This allows the LSMT to operate on a representation that is abstracted away from the (sensory) input and solve the reasoning tasks that require extrapolation over input symbols.

Pros:
- This is interesting work that addresses an important question in neuroscience and deep learning.
- The tasks proposed here can be used to evaluate the other models

Cons:
- They test the model only on synthetic tasks
- The model feels ad-hoc for the task. It is unclear if the model can be useful for other more complex problems.
- I believe the confidence score is crucial as it tells LSTM whether the same object was already in the memory. Some ablation on this is useful. Specially, the confidence score seems to provide a quite reliable high-level feature for the LSTM controller, which is not necessarily a weakness but providing ablation could be nice.

Additional comments:
- Curious to see how more recent associative MANNs perform on these tasks, such as TPR-RNN (https://github.com/ischlag/TPR-RNN) and MNM (https://bitbucket.org/tsendeemts/mnm/src/master/). Since the code is already out there, it shouldn’t be too difficult to include them as baseline.
- Smolensky’s TPR, which is one of the earliest works on the topic is currently missing from the discussion

---

> ### Author Response · Authors · 2020-11-23
> **Reply to Reviewer 3**
>
> We would like to thank the reviewer for the useful suggestions and thoughtful feedback on our work. The reviewer noted that, in the original manuscript, we did not evaluate whether the confidence values appended to retrieved memories are an important component of the ESBN. We agreed that this is important to evaluate, and have now included ablation experiments to better understand this (summarized in paragraph 5 of section 4, and described in detail in section A.4.5 of the revised manuscript). These experiments revealed that the confidence values were helpful to the ESBN in the same/different and RMTS tasks, but not in the distribution-of-three or identity rules tasks. This makes sense given that the former, but not the latter, require an evaluation of whether the current percept is already in memory (which can be made using a familiarity judgement), whereas the latter require the retrieval of a particular memory (i.e., the best match to the current percept). In the revised manuscript, we relate this dissociation to the distinction in cognitive psychology between familiarity and recollection [1]. We also discuss (in section A.4.5) an alternative approach that implicitly gives the model access to the same information necessary to compute confidence values. This is accomplished by initializing memory at the beginning of each sequence with a learned ‘default’ entry (as a proxy for information already present in episodic memory) against which incoming percepts can be compared. We show that this approach is about as effective as including explicit confidence values. We believe that these results and discussion enrich the paper, and thank the reviewer for suggesting this experiment.
>
> The reviewer expressed concern about the synthetic nature of the tasks that we studied, and expressed doubt about whether the ESBN model would be as effective at more complex tasks. We agree that an important next step will be to evaluate the ESBN on more complex tasks. We argue however that the current tasks, though synthetic, successfully capture  -- and allowed us to isolate and focus on -- the key property of out-of-distribution generalization that we aimed to test, as evidenced by the fact that a range of competitive baselines failed to generalize well on these tasks. Additionally, we would like to emphasize that we chose to focus on relatively simple tasks because doing so made it easier to understand as much as possible about how and why the ESBN operates as it does, and to evaluate our central claim that the ESBN learns symbol-like representations. Nevertheless, we agree that testing the ESBN on more complex tasks is an important next step, and intend to do so in future work.

---

> > ### Author Response · Authors · 2020-11-23
> > **(continued)**
> >
> > The reviewer suggested that we evaluate  other recent external memory models on our tasks, and suggested two specific candidate models. The first suggested model, the TPR-RNN [2], is a very interesting approach but, as currently formulated, assumes a nested temporal structure (since it was developed to operate over a sequence of sentences) and so cannot be straightforwardly applied to our tasks. We believe some careful thought and experimentation is necessary to fairly apply this model to our tasks, and consider this to be another important next step in our work. The second suggested model, MNM [3] was much more straightforward to apply ‘out of the box’ to our tasks. We found that MNM performed about as well as the NTM on our tasks, and have included these results in the revised manuscript (section 4). We note however that one of the implications of our work is that the specific nature of the external memory mechanism (so long as it supports some form of arbitrary binding) may be independent of the two-stream architectural structure of the ESBN, that we propose is a critical component of its success in generating symbolic representations. This same architectural structure could, in principle, be combined with other external memory mechanisms, such as the tensor product fast weights employed in the TPR-RNN, or the learned mapping function in MNM. These possibilities are certainly interesting directions for future work, and we thank the reviewer for bringing these alternative approaches to our attention.
> >
> > Finally, the reviewer notes that we neglected to mention the tensor product representation [4] in the discussion section of the initial manuscript. This was an egregious oversight on our part. We have now included a discussion of this approach, and how it relates to our proposed approach, in the discussion (section 6.2).
> >
> > [1] Yonelinas, A. P. (2001). Consciousness, control, and confidence: the 3 Cs of recognition memory. Journal of Experimental Psychology: General, 130(3), 361.
> >
> > [2] Schlag, I., & Schmidhuber, J. (2018). Learning to reason with third order tensor products. In Advances in neural information processing systems (pp. 9981-9993).
> >
> > [3] Munkhdalai, T., Sordoni, A., Wang, T., & Trischler, A. (2019). Metalearned neural memory. In Advances in Neural Information Processing Systems (pp. 13331-13342).
> >
> > [4] Smolensky, P. (1990). Tensor product variable binding and the representation of symbolic structures in connectionist systems. Artificial intelligence, 46(1-2), 159-216.

---

### Official Review · AnonReviewer1 · 2020-10-29
**Decoupling symbol-like mechanism from concrete entities**

**Rating:** 7
**Confidence:** 4

**Review:**

Summary:
This paper addresses abstract rule learning in high-dimensional data through constructing a recurrent neural network that exhibits a variable-binding ability. The proposed method, ESBN, is a RNN augmented with two memories, one for keys and one for values. The key memory captures the relations between items, which are important to produce symbolic answer (e.g.,  item/group index). The value memory stores the item embedding. Tested on several abstract rule learning tasks on visual inputs, ESBN exhibits excellent generalization with limited training data.

Pros:
- The problem of learning abstract rules that are decoupled from actual content (visual symbols in this case) is important, and is currently under-studied in machine learning.
- The solution, ESBN, is simple and effective. Empirical evaluations clearly show that ESBN woks well on tested tasks, while several well-studied neural  networks (LSTM, NTM, Relational network, Transformers) fail.

Cons:
- While the results are very encouraging on the simplified tasks, we still do not really know WHY the model works the way it does. After all, the relation between keys are transferred from the similarity between corresponding values. The tasks tested seem to emphasize on the object matching, and thus may not work on other types of rules, or with more variations of inputs.
- The paper is motivated well and the high-level idea is described, but certain important algorithmic/design details are missing in the main text (e.g., the role of confidence, gate). Algorithm 1 is at core of ESBN, but is left to Appendix, making it hard to follow the unrolling of the controller in step t. The concatenation of the key memory with the confidence score seems to make the dimension of k_r grows with t. Is this intended? If not, then how a fixed size input to the RNN controller is implemented?

Others:
- The paper conjectures that that the poor result of the Relation Network (RN) is due to its focus on pair-wise relations. More recent follow-ups have introduced higher-order relations (e.g., Temporal Relation Network [1] and Conditional Relation Network [2]) may be able to mitigate the issue. It would be nice to have these tested, or discussed in the context of abstract rules learning.
-  It would be important to explain/discuss the effect of temporal context normalization on ESBN.


Reference:
[1] Bolei Zhou, Alex Andonian, Aude Oliva, and Antonio Torralba. "Temporal relational reasoning in videos". In Proceedings ofthe European Conference on Computer Vision (ECCV), pages 803–818, 2018.
[2] Le, Thao Minh, et al. "Hierarchical Conditional Relation Networks for Video Question Answering." Proceedings of the IEEE/CVF Conference on Computer Vision and Pattern Recognition. 2020.

---

> ### Author Response · Authors · 2020-11-23
> **Reply to Reviewer 1**
>
> We would like to thank the reviewer for the helpful suggestions of how to improve upon our work. The reviewer notes that the original manuscript would benefit from a clearer explanation of why and how the ESBN actually works. We agreed, and have now included some additional discussion and analyses to address this concern (paragraphs 4-6 of section 4). Specifically, we now include a consideration of the following points:
> - We perform experiments to test the extent to which the ESBN depends on the use of a convolutional encoder. These experiments showed that the ESBN performs equally well with the use of either an MLP or random projection as encoders. This result strengthens our conclusion that the ESBN is performing something close to abstract variable-binding, since it shows that the ESBN can perform the task well with any arbitrary set of entities, regardless of how they are encoded.
> - We perform an ablation experiment to determine the importance of the confidence values used by the ESBN. This reveals that these confidence values are important for some tasks but not others, which we point out are consistent with work in the cognitive psychology of memory (we also elaborate on this point in response to Reviewer 3 below).
> - The initial manuscript included analyses, in the appendix, of the learned key representations in the ESBN, showing that they have a symbol-like structure. We now include a paragraph in the main body of the manuscript summarizing these results.
>
> We’d like to thank the reviewer for noting this decificiency in the original manuscript. We believe the revised manuscript is significantly improved by these additional analyses and discussion.
>
> The reviewer suggested that the algorithm describing the ESBN should be moved to the main body of the manuscript. We agreed and did so. The algorithm, figure, and textual description of the model are now all in the same location in the manuscript (section 3.2). We hope that this will make it easier for the reader to understand how the model works. To address the reviewer’s specific question about the way in which confidence values are appended: $k_{r_{t}}$ is formed by concatenating the retrieved key from the current time step, and the confidence value for the current time step ($c_{k_{t}}$). The confidence values from previous time steps are not included, so $k_{r_{t}}$ does not grow with $t$. We have clarified this in the textual description by specifying that only $c_{k_{t}}$ is appended to the retrieved key.
>
> The reviewer directed us to recent work on relation nets that introduces novel methods for incorporating n-ary relations. We tested one of these models (detailed implementation described in section A.2.8), the Temporal Relation Network [1], on the distribution-of-three and identity rules tasks. This did improve performance somewhat over the standard RN (which is now mentioned in paragraph 2 of section 4), but not as much as training the standard RN on a larger training set. In the appendix (section A.4.2), we discuss a few potential reasons why this wasn’t as effective on these particular tasks as might have been expected.
>
> The reviewer also notes that the manuscript would benefit from more discussion of the effect of temporal context normalization. We agreed, and included a brief explanation of the rationale behind temporal context normalization in section 3.1 of the revised manuscript. We thank the reviewer for pointing this out.
>
> [1] Bolei Zhou, Alex Andonian, Aude Oliva, and Antonio Torralba. "Temporal relational reasoning in videos". In Proceedings of the European Conference on Computer Vision (ECCV), pages 803–818, 2018.

---

### Decision · Program_Chairs · 2021-01-07
**Final Decision**

**Decision:**

Accept (Spotlight)

**Comment:**

The paper proposes a recurrent neural network architecture for abstract rule learning. An LSTM is augmented with a two-stream memory structure: one block is populated with visual representations, and the other is populated by hidden state vectors from the RNN controller.

The authors also introduce a set of tasks that require a simple symbolic reasoning on visual inputs and strong extrapolation ability. They show that previous memory-augmented neural networks fail on these tasks, whereas their model exhibits excellent generalization with limited training data.

Pro: The work addresses an important and open question in neuroscience and deep learning. The proposed solution is simple and effective. The manuscript is well-written. It was also improved in a revised version after the first review round.

Con: The main criticism raised by the reviewers was that the considered tasks may be a too simplified synthetic task.  It would have been good to consider other the more complex tasks involving symbolic reasoning such as CLEVR or bAbI.

While this is a valid criticism, all reviewers agreed that this is an interesting and important work worth publishing. In particular, the considered question is of pivotal importance for the community and the work presents a significant progress.